March 29, 2021

# A discontinuous Galerkin finite element model for fast channelized lava flows v1.0

Colton J. Conroy[1,2] and Einat Lev[1]

Lamont-Doherty Earth Observatory[1]
Columbia University in the City of New York, USA

Roy M. Huffington Department of Earth Sciences[2]
Southern Methodist University, Dallas, TX, USA

## Abstract

Lava flows present a significant natural hazard to communities around volcanoes and are typically slow moving ($< 1$ to 5 cm/s) and laminar. Recent lava flows during the 2018 eruption of Kilauea Volcano, Hawai'i, however, reached speeds as high as 11 m/s and were transitional to turbulent. The Kilauea flows formed a complex network of braided channels departing from the classic rectangular channel geometry often employed by lava flow models. To investigate these extreme dynamics we develop a new lava flow model that incorporates nonlinear advection as well as a nonlinear expression for the fluid viscosity. The model makes use of novel discontinuous Galerkin (DG) finite element methods and resolves complex channel geometry through the use of unstructured triangular meshes. We verify the model against an analytic test case and demonstrate convergence rates of $\mathcal{P} + 1/2$ for polynomials of degree $\mathcal{P}$. Direct observations recorded by Unoccupied Aerial Systems (UASs) during the Kilauea eruption provide inlet conditions, constrain input parameters, and serve as a benchmark for model evaluation.

## 1   Introduction

On May 3, 2018, Kilauea Volcano on the Island of Hawai'i began to erupt from new fissures in the lower East Rift Zone at the center of the Leilani Estates Subdivision. Before ceasing in early August 2018, the lava flows destroyed over 650 structures and caused significant damage to infrastructure and essential facilities. During the second half of the eruption the flow field established a complex braided channel system (which is common to many basaltic flows), originating from Fissure number 8 (see Figure 1). The "Fissure 8" flows were unique in the fact that they produced channelized flows reaching speeds as high as 15 m/s (Patrick et al., 2019). These high speeds, coupled with channel geometry (e.g. constrictions) produced Reynolds numbers ($Re > 3000$) that were significantly higher than typical lava flows. To investigate these extreme dynamics we develop a new channelized lava flow computer model named a discontinuous Galerkin finite element model for fast channelized lava flows version 1.0.

This paper is organized as follows: in §1.1–§1.3 we present the motivation for this work, as well as background on the mathematical tools we employ. in §2 we present the mathematical

model along with the bottom stress calculation and detail its nuances. We present the DG numerical discretization of the mathematical model in §3 and verify the model in §4. In §5 we evaluate the model against observations of lava flows from the 2018 eruption of Kilauea volcano. We present misfit errors and root mean square (RMS) errors for the velocity field from a braided channel section of Fissure 8, and provide quantitative insight into physical quantities of the lava flow field in this area including its thickness and viscosity. We close the paper in §6 with some discussion and conclusions.

## 1.1  Motivation

Typical "operational" lava flow models simulate unconfined lava flow in a 2D plan view [e.g., SCIARA (Crisci et al., 2004-04), MAGFLOW (Vicari et al., 2007), LavaPL (Connor et al., 2012), VOLCFLOW (Kelfoun and Vargas, 2015)] using either cellular automata or depth-averaged equations in an effort to forecast the area of land inundated by the lava. It is often difficult, however, for these models to accurately reproduce the complicated braided channel network such as those created by "Fissure 8." These braided channel networks are common in natural flows (e.g., Dietterich and Cashman, 2014-08) and understanding the evolution of the velocity, rheology, and temperature fields (e.g. in response to pulsating effusion) within these channels is critical to hazard mitigation (Patrick et al., 2019). Direct measurements of lava properties in situ is usually extremely difficult and dangerous. Modeling lava dynamics within the bounds of an established channel can help to better understand material properties of the flowing lava and inform models and decisions.

Previous attempts to model channelized lava flows have made use of simple heuristic formulas such as Jeffreys equation for laminar flows (Harris and Rowland, 2015) or Chezy approximations for higher speed flows (Baloga et al., 1995). While convenient, the use of these equations has largely been dictated by the fact that it has been difficult to obtain the physical data necessary for advanced modeling efforts (e.g. channel domain boundaries, inlet boundary conditions, topography, etc). However, with the advent of Unoccupied Aerial Systems (UASs, or 'drones') and their ability to survey active lava fields, we now have access to the data required by sophisticated numerical methods.

## 1.2  Shallow-water equations for fast lava flows

Commensurate with this development in observational capabilities, we introduce a numerical method for modeling fast moving lava flows in complex channels. The high Reynolds number associated with these lava flows coupled with the fact that the total length of the flows (on the order of kilometers) is much greater than the flow depth (on the order of meters) means that the dynamics can be well approximated by two-dimensional depth-integrated equations for mass, momentum, and energy. In particular, we utilize a system of dynamical equations known as the *shallow water equations* (De Saint Venant (1864) and Boussinesq (1872)), which quantify average horizontal velocities and the depth of flow. These equations are traditionally used to model free surface flows in coastal oceanic regions, estuaries, and rivers (Dawson and Mirabito, 2008), although they have been used to model debris flows (George and Iverson, 2014) and lava flows (Costa and Macedonio, 2005) as well. The main assumption in the shallow water theory is that the fluid pressure is hydrostatic; gravitational acceleration

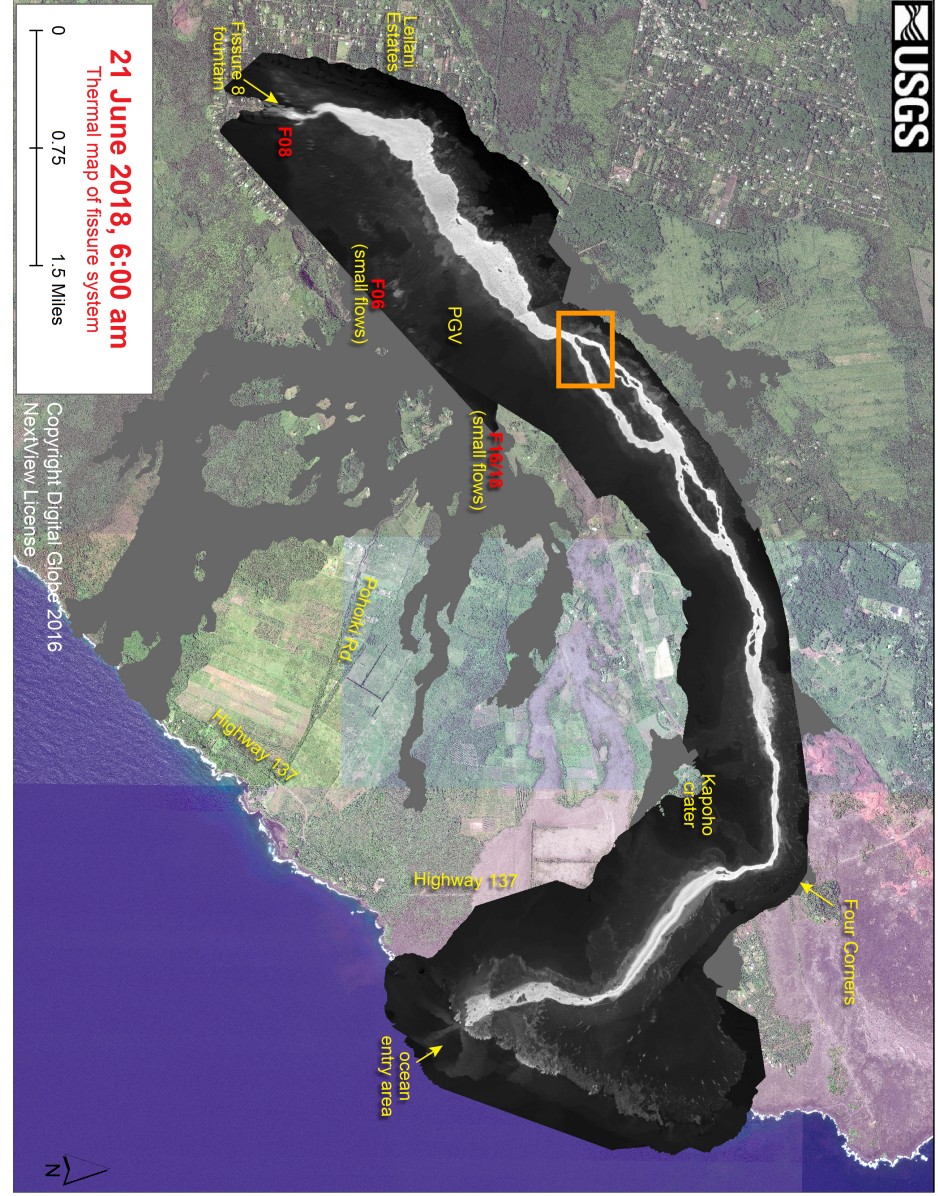

Figure 1: A satellite image (colored, in the background, by DigitalGlobe) overlaid by a thermal aerial orthomosaic (grayscale) where the white and light gray areas reveal the path of the Fissure 8 flow channel as it was on June 21st, 2018. Data and map by USGS. The orange rectangle depicts the area of UAS site 8, from where the video we analyzed was captured on June 22nd, 2018. The flat gray areas south of the active flow channel demarcate the areas inundated by lava during the early stages of the eruption. North is up. PGV is the Puna Geothermal Ventures power plant that was heavily impacted by the lava.

in the fluid dominates vertical accelerations and the pressure is calculated via the vertical momentum equation. The formulation of the shallow water equations that we utilize is designed specifically for advection dominated flows (Kubatko et al., 2006) and the pressure gradient term is formulated so that the dynamical equations are well balanced; steady states are preserved and no artificial motion is induced by numerical artifacts (see Conroy (2014) for a full derivation of the dynamical equations from conservation principles).

Lava flows are distinct from hydrological free surface flows in the sense that lava transfers heat to its surroundings; as lava effuses from a vent it cools along lateral flow boundaries and can form solid walls ('levees') that prevent the lava from spreading to nearby regions. If lava effusion extends for several days, long channels may form that efficiently transport lava from the vent to the flow front. The speed at which the lava flows through the channel system depends on the viscosity of the lava, which in turn is highly dependent on the temperature and chemical composition of the lava (e.g., Griffiths, 2000). The presence of crystals and/or bubbles in the lava can make its viscosity non-Newtonian (Manga et al., 1998; Mader et al., 2013) and thus strongly dependent on stress gradients and the thermal properties of the lava. To reflect this strong dependence on temperature, we solve a depth-integrated energy equation that quantifies the thermal evolution of the lava as it interacts with its environment. The depth-integrated energy equation is coupled to the shallow water equations through a thermally dependent non-linear stress term that reflects the rheology of the lava, and can account for the presence of crystals and/or bubbles in the lava flow.

The logistical key to using shallow water equations to model lava flow dynamics rests on the development of the non-Newtonian bottom stress term. Typical friction drag laws do not take into account the viscosity of the fluid (due to the assumption that the fluids inertial acceleration is much greater than its internal resistance). However, in our particular case the flow is not fully turbulent; internal resistance needs to be taken into account in some fashion. Thus, we express the stress at the bottom boundary as a function of the temperature and the vertical stress gradient (which is a function on the vorticity). We solve a thermal boundary layer problem to calculate the temperature at the bottom boundary and utilize the vorticity to determine a virtual length scale over which the interior velocity goes from the depth-averaged value to zero. This results in a bottom stress approximation that is void of a friction factor (e.g. Manning's $n$) and allows scientists to study physical properties of the lava that are difficult to measure directly (e.g., viscosity). For example, application of the model to the Kilauea 2018 Fissure 8 lava flows reveals that the lava behaved as a shear thickening fluid due to the high bubble content ($\approx 50\%$) with a large capillary number and agree well with the recent lab experiments of Lev et al. (2020).

## 1.3 The Discontinuous Galerkin Finite Element Method

Because closed form analytic solutions do not exist to the nonlinear shallow water equations and energy equation, we construct approximate solutions to these equations using discontinuous Galerkin (DG) finite element methods (Cockburn and Shu, 2001), which have been used to successfully model other geophysical fluid flows including coastal ocean circulation (Kärnä et al., 2018), hurricanes (Dawson et al., 2011), avalanches (Patra et al., 2006), and debris flows (Conroy and George, 2020). The DG finite element method differs from continuous Galerkin (CG) finite element methods in that the DG method solves an integral (or

weak) form of the mathematical equations over *individual* elements and utilizes a solution space that is discontinuous across element boundaries. This allows the DG method to resolve steep gradients that form in the numerical solution, such as the thermodynamic gradients that form at lava channel wall boundaries. Even though the DG method is discontinuous it still conserves mass, momentum, and energy both locally and globally by utilizing a numerical flux function (introduced by finite volume methods, see LeVeque (2002) for instance) that takes the discontinuous state of physical properties at element boundaries and creates a consistent flow of information from element-to-element (Cockburn and Shu, 2001).

Further, the DG method has been shown to be highly parallelizable using high performance computing (e.g. Kubatko et al. (2009) and Patra et al. (2006)) and it is amenable to unstructured numerical meshes. The later feature is important when resolving geometrically complex boundaries of a given fluid domain such as the flow fields commonly produced by basaltic flows. For instance, the lava flows that effused from Fissure 8 formed a complicated network of braided channels, with multiple locations of branching and merging. In addition, obstructions such as large lava rafts or preexisting structures caused local disruptions in the flow field, making it difficult to evaluate the dynamics using a simplified one-dimensional channelized model, such as those that use a classical rectangular channel geometry (Harris and Rowland, 2015). To account for these complexities, our new lava flow model discretizes the lava channel domain with an unstructured triangular mesh. This reduces model error as it pertains to a representation of the lava flow domain and is important when reproducing localized flow features of the lava field, such as the jet visible in Figure 4.

Model verification consists of solving an analytic test case using forcing functions that we choose to exactly satisfy the equations of motion, and results indicate that for smooth solutions the method converges to the exact solution at a rate of $\mathcal{P} + 1/2$ for polynomials of degree $\mathcal{P}$.

# 2   Mathematical model

Fluid flow on a sloped terrain can be quantified in a Cartesian-coordinate $(x, y, z)$ system over a time dependent domain $\Omega(t) \in \mathbb{R}^3$ by solving Eulerian conservation equations of mass, linear momentum, and energy,

$$\frac{\partial \rho}{\partial t} \; + \; \nabla \cdot (\rho \mathbf{u}) \;\; = \;\; 0, \tag{1}$$

$$\frac{\partial}{\partial t}(\rho \mathbf{u}) + \rho \mathbf{u} \cdot \nabla \mathbf{u} \; + \; \nabla p \; - \; \left(\nabla' \boldsymbol{\tau}\right)' \;\; = \;\; \mathbf{f}_b, \tag{2}$$

$$\rho c_p \frac{\partial T}{\partial t} \; + \; \rho c_p \left(\mathbf{u} \cdot \nabla T\right) - \nabla \cdot \mathbf{q} \;\; = \;\; \dot{q}. \tag{3}$$

In equations (1)–(3), $\rho$ is the fluid density, $\mathbf{u} = (u, v, w)'$ is the fluid velocity vector, $\nabla = (\frac{\partial}{\partial x}, \frac{\partial}{\partial y}, \frac{\partial}{\partial z})'$ is the gradient operator, $p$ is the pressure, and $\boldsymbol{\tau}$ is the stress term,

$$\boldsymbol{\tau} \;=\; \begin{bmatrix} \tau_{xx} & \tau_{xy} & \tau_{xz} \\ \tau_{yx} & \tau_{yy} & \tau_{yz} \\ \tau_{zx} & \tau_{zy} & \tau_{zz} \end{bmatrix}. \tag{4}$$

We denote the x-component of gravitational acceleration that is tangential to the sloped surface by $g_x$; $g_y$ is the y-component of gravitational acceleration that is tangential to the sloped surface, and $g_z$ is gravitational acceleration that is normal to the sloped surface. Collectively, these terms form the body force vector $\mathbf{f}_b = (g_x,\, g_y,\, g_z)'$ acting on the fluid. $T$ is the temperature of the fluid, $c_p$ is the specific heat capacity of the fluid, $\mathbf{q} = (k_T \frac{\partial T}{\partial x},\, k_T \frac{\partial T}{\partial y},\, k_T \frac{\partial T}{\partial z})'$ is the heat flux through the fluid ($k_T$ is a conduction heat transfer coefficient that measures the spread of heat within the fluid), and $\dot{q}$ quantifies the generation/dissipation of heat within the fluid.

quations (1)and (2), taken together, are the Navier-Stokes equations and quantify the force dynamics acting on the fluid (see Conroy (2014) for a derivation from first principles). Equation (3) is the thermal energy equation; it quantifies the transport of energy through the fluid due to internal temperature gradients and differences between the fluid temperature and the temperature of the surrounding medium (see Moran et al. (2003) for instance). To apply equations (1)–(4) to channelized lava flow we need to supplement them with appropriate boundary conditions and define the stress matrix $\boldsymbol{\tau}$. Here, we assume that the system of equations given by (1)–(3) are subject to the following kinematic, dynamic, and thermal boundary conditions:

- Channel wall boundary condition:

$$\begin{aligned} \text{no normal flow,} \qquad \mathbf{u} \cdot \hat{\mathbf{n}} &= 0, \\[4pt] \text{no pressure gradient,} \qquad \partial p / \partial \hat{\mathbf{n}} &= 0, \\[4pt] \text{slip velocity,} \qquad \mathbf{u} \cdot \hat{\mathbf{t}} &= f(\boldsymbol{\tau}_{\text{wall}}, \boldsymbol{\sigma}_{\text{wall}}), \\[4pt] \text{heat loss via conduction,} \qquad \mathbf{q} \cdot \hat{\mathbf{n}} &= (k_T / h_w)\,(T - T_{\text{wall}}). \end{aligned}$$

- Inlet boundary condition:

$$\begin{aligned} \text{prescribed velocity,} \qquad \mathbf{u} \cdot \hat{\mathbf{n}} &= \text{prescribed}, \\[4pt] \text{prescribed pressure,} \qquad p &= \text{prescribed}, \\[4pt] \text{prescribed heat content,} \qquad \rho c_p T &= \text{prescribed}. \end{aligned}$$

- Outlet boundary condition:

$$\begin{aligned} \text{zero change in normal velocity,} \qquad \partial \mathbf{u} / \partial \hat{\mathbf{n}} &= 0, \\[4pt] \text{zero change in pressure,} \qquad \partial p / \partial \hat{\mathbf{n}} &= 0, \\[4pt] \text{zero change in heat content,} \qquad \rho c_p \partial T / \partial \hat{\mathbf{n}} &= 0. \end{aligned}$$

- Free surface boundary condition at $z = \zeta$:

  no relative normal flow, $\qquad\qquad\qquad\qquad \partial\zeta/\partial t = -u(\partial\zeta/\partial x) - v\partial(\zeta/\partial y) + w,$

  atmospheric pressure, $\qquad\qquad\qquad\qquad p = p_{\mathrm{atm}},$

  surface shear stress, $\qquad\qquad\qquad\qquad \tau_{sx} = -\tau_{xx}(\partial\zeta/\partial x) - \tau_{yx}(\partial\zeta/\partial y) + \tau_{zx},$

  $\qquad\qquad\qquad\qquad\qquad\qquad\qquad\quad \tau_{sy} = -\tau_{xy}(\partial\zeta/\partial x) - \tau_{yy}(\partial\zeta/\partial y) + \tau_{zy},$

  heat loss via radiation and convection, $\quad \mathbf{q} \cdot \hat{\mathbf{n}} = \epsilon\sigma_B\left(T^4 - T_{\mathrm{atm}}^4\right) + k_c\left(T - T_{\mathrm{atm}}\right)^{\frac{4}{3}}.$

- Bottom boundary condition at $z = -h$:

  no slip velocity, $\qquad\qquad\quad \mathbf{u} = 0,$

  bottom shear stress, $\qquad\quad \tau_{bx} = \tau_{xx}(\partial h/\partial x) + \tau_{yx}(\partial h/\partial y) + \tau_{zx},$

  $\qquad\qquad\qquad\qquad\qquad \tau_{by} = \tau_{xy}(\partial h/\partial x) + \tau_{yy}(\partial h/\partial y) + \tau_{zy},$

  heat loss via conduction, $\quad \mathbf{q} \cdot \hat{\mathbf{n}} = \left(k_b/h_b\right)\left(T - T_{\mathrm{ground}}\right).$

In the boundary conditions above, $\hat{\mathbf{n}}$ is the unit normal vector to the wall boundary, outlet boundary, inlet boundary, moving free surface, and bottom boundary, respectively; $\hat{\mathbf{t}}$ is the unit tangential vector to the wall, $\boldsymbol{\tau}_{\mathrm{wall}}$ is the tangential shear stress at the wall, and $\boldsymbol{\sigma}_{\mathrm{wall}}$ is the normal stress at the wall. We denote the temperature at the wall by $T_w$, $k_w$ is the thermal conductivity of the wall, and $h_w$ is the thickness of the thermal boundary layer through which heat is conducted from the lava flow to the wall. We measure the free surface, $\zeta$, relative to a steady depth of flow, $h(x,y)$, that serves as the zero datum in the $z$-direction (see Figure 2). The surface forces acting on the free surface consist of the atmospheric pressure, $p_{\mathrm{atm}}$, along with a surface shear stress applied by the wind $\boldsymbol{\tau}_s = (\tau_{sx},\,\tau_{sy})'$. Heat transfer from the lava surface to the surrounding atmosphere is dominantly due to radiation and air convection where $\epsilon$ is the emissivity of the lava, $\sigma_B$ is the Stefan-Boltzmann constant, $T_{\mathrm{atm}}$ is the temperature of the surrounding atmosphere, and $k_c$ is the convection heat transfer coefficient. The main resisting force in dense shallow mass high speed flows comes from the bottom stress, $\boldsymbol{\tau}_b = (\tau_{bx},\,\tau_{by})'$, which is a function of the temperature of the lava at the basal boundary, where $T_{\mathrm{ground}}$ is the temperature of the ground and $h_b$ is the depth of the thermal boundary layer through through which heat is transferred from the lava to the ground.

Theoretically, we could define $\boldsymbol{\tau}$ and solve equations (1)–(3) along with the prescribed boundary conditions to model channelized lava flows. In practice, however, we need to simplify equations (1)–(3) to make the solution more tractable. More specifically, we assume the following: i. the lava flow field is incompressible, ii. vertical accelerations in the lava are dominated by gravity, iii. lava flow lengths are much greater than the flow depth and horizontal flow speeds are large enough that stress gradients are dominated by the first two columns of the stress matrix $\boldsymbol{\tau}$.

Assumption i. reduces the conservation of mass equation to $\nabla \cdot \mathbf{u} = 0$, while assumptions ii. and iii. reduce the $z$-momentum equation to,

$$\frac{\partial p}{\partial z} = \rho g_z,$$

which we can leverage to determine the pressure. Integrating from the free surface $\zeta$ down to a given $z$-coordinate yields,

$$p = p_{\text{atm}} + \rho g_z \left( \zeta - z \right) \tag{5}$$

We assume that gradients in $p_{atm}$ are negligible and the horizontal pressure gradient in equation (2) becomes,

$$\bar{\nabla} p = \bar{\nabla}(\rho g_z \zeta),$$

where $\bar{\nabla} = (\frac{\partial}{\partial x}, \frac{\partial}{\partial y})'$. We further simplify the mathematical model and eliminate the vertical dimension by integrating $\nabla \cdot \mathbf{u} = 0$ and equations (2) and (3) over the depth of the lava flow (from $-h$ to $\zeta$). We then apply Leibniz's integral rule, utilize the free-surface and bottom boundary conditions, assume the density is constant, and simplify the resulting expression to arrive at the following depth-integrated equations,

$$\frac{\partial H}{\partial t} + \frac{\partial H\bar{u}}{\partial x} + \frac{\partial H\bar{v}}{\partial y} = 0, \tag{6}$$

$$\frac{\partial H\bar{u}}{\partial t} + \frac{\partial}{\partial x}\left(H\bar{u}^2\right) + \frac{\partial}{\partial y}\left(H\bar{u}\bar{v}\right) + H\frac{\partial}{\partial x}\left(g_z\zeta\right) + Hg_x = \frac{1}{\rho}\left(\tau_{sx} - \tau_{bx} + f_x\right), \tag{7}$$

$$\frac{\partial H\bar{v}}{\partial t} + \frac{\partial}{\partial x}\left(H\bar{u}\bar{v}\right) + \frac{\partial}{\partial y}\left(H\bar{v}^2\right) + H\frac{\partial}{\partial y}\left(g_z\zeta\right) + Hg_y = \frac{1}{\rho}\left(\tau_{sy} - \tau_{bx} + f_y\right), \tag{8}$$

$$\frac{\partial H\bar{T}}{\partial t} + \frac{\partial H\bar{u}\bar{T}}{\partial x} + \frac{\partial H\bar{v}\bar{T}}{\partial y} = q_s + q_b + \bar{\nabla} \cdot \bar{q}_i + \dot{q}, \tag{9}$$

where $f_x$ and $f_y$ are the x-component and y-component of the depth-averaged gradient of the shear stresses acting on vertical fluid planes, $q_s$ is the heat flux through the free surface, $q_b$ is the heat flux at the bottom boundary, $\bar{\nabla} \cdot \bar{q}_i$ quantifies depth-averaged internal conduction, and $\dot{q}$ represents depth-averaged internal heat generation/dissipation. The depth-averaged velocity, $\bar{\mathbf{u}} = (\bar{u}, \bar{v})$, and depth-averaged temperature, $\bar{T}$, are defined as,

$$\bar{u} = \frac{1}{H}\int_{-h}^{\zeta} u \, \mathrm{d}z, \tag{10}$$

$$\bar{v} = \frac{1}{H}\int_{-h}^{\zeta} v \, \mathrm{d}z, \tag{11}$$

$$\bar{T} = \frac{1}{H}\int_{-h}^{\zeta} T \, \mathrm{d}z. \tag{12}$$

The above system of depth-integrated equations (6)–(9) can be further simplified due to the dynamics of high speed flows. More specifically, $f_x$ and $f_y$ are negligible in high speed flows except at no slip and small slip velocity boundary conditions where large stress gradients form due to the decay of the velocity field to a value of zero (or near zero). In our quantitative analysis of UAS footage from the 2018 Kilauea eruption, lava flow velocities at channel wall boundaries were much greater than zero, and therefore, we utilize a slip (no flow) channel wall boundary condition and neglect $f_x$ and $f_y$ in this initial version of the model (see Rao and Rajagopai (1999) for an in depth investigation on channel wall boundary conditions in terms of the slip versus no slip condition). The surface stress terms $(\tau_x, \tau_y)'$ in the depth-integrated equations account for wind stress on the lava flow, which we assume to be negligible due to the ratio of the density of air to the density of lava being much less than one. We include the effect of heat conduction at the channel wall boundaries, but we neglect heat conduction in the interior of the lava due to the high speed of the flow and we set $\bar{\nabla} \cdot \bar{q}_i = 0$. Internal heat generation/dissipation can be significant in lava flows with a high crystal content (Griffiths, 2000) and in lava flows in closed tubes (Costa and Macedonio, 2005). The fissure 8 lava flows were hot with limited crust cover and samples indicate that the crystal content was low in the channel section that we apply the model to (Gansecki et al., 2019), and therefore, we neglect $\dot{\bar{q}}$ in the current model but plan to include it in future releases.

It can be noted that the pressure gradient terms, $(1/\rho)\bar{\nabla}p = (H\frac{\partial}{\partial x}(g_z\zeta), H\frac{\partial}{\partial x}(g_z\zeta))'$, in equations (7) and (8) are non-conservative product terms that can lead to entropy violating numerical fluxes if care is not taken in evaluating them numerically (see LeVeque (2002), for instance). To circumvent this issue, we make use of the fact, $H(x, y, t) = \zeta(x, y, t) + h(x, y)$, and re-write equations (7) and (8) in the conservative form (Kubatko et al. (2006)),

$$\frac{\partial \zeta}{\partial t} + \frac{\partial H\bar{u}}{\partial x} + \frac{\partial H\bar{v}}{\partial y} = 0, \tag{13}$$

$$\frac{\partial H\bar{u}}{\partial t} + \frac{\partial}{\partial x}\left(H\bar{u}^2 + P\right) + \frac{\partial}{\partial y}(H\bar{u}\bar{v}) + Hg_x = g\zeta\frac{\partial h}{\partial x} - \frac{\tau_{bx}}{\rho}, \tag{14}$$

$$\frac{\partial H\bar{v}}{\partial t} + \frac{\partial}{\partial x}(H\bar{u}\bar{v}) + \frac{\partial}{\partial y}\left(H\bar{v}^2 + P\right) + Hg_y = g\zeta\frac{\partial h}{\partial y} - \frac{\tau_{by}}{\rho}, \tag{15}$$

$$\frac{\partial H\bar{T}}{\partial t} + \frac{\partial H\bar{u}\bar{T}}{\partial x} + \frac{\partial H\bar{v}\bar{T}}{\partial y} = q_s + q_b, \tag{16}$$

$$\tag{17}$$

where $P = \frac{1}{2}g_z\left(H^2 - h^2\right)$ is the pressure flux and $\partial h/\partial x$ and $\partial h/\partial y$ quantify the gradient in the steady reference depth of flow that $\zeta$ is measured relative to. We supplement the system of equations given by (17) with initial conditions along with the channel wall, inlet, and outlet boundary conditions. It can be noted that the depth-integrated mass and momentum equations given by equations (13)–(15) are well studied in the literature and are commonly used to model shallow mass flows such as coastal ocean circulation and hurricane storm surge, see for example Dawson et al. (2011) and Kubatko et al. (2006). The addition of the energy equation complicates the solution of equations (14) and (15) due to the fact that the

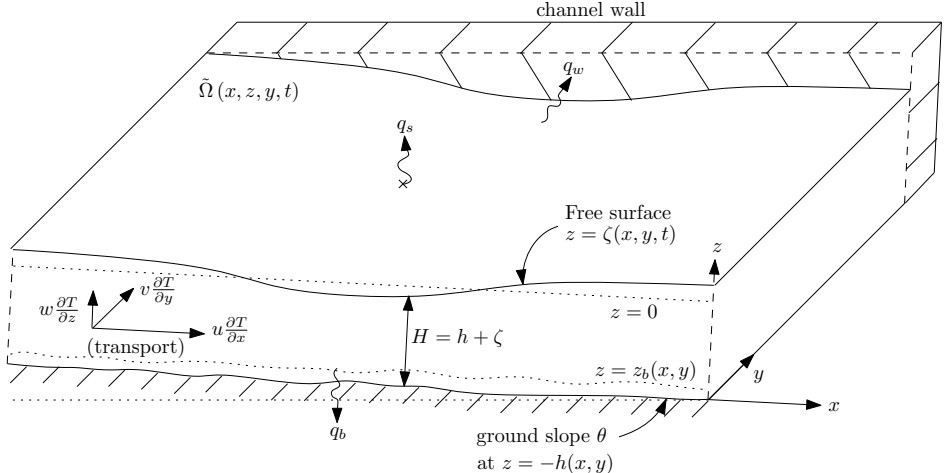

Figure 2: A vertical cross-section along the center line of the flow, showing the coordinates ($x$ and $z$, with $y$ the across-flow direction) and the heat transfer mechanisms considered in the model (conduction to the base, advection by the flow, and heat loss by radiation/convection at the surface).

bottom stress term, $\boldsymbol{\tau}_b = (\tau_{bx},\, \tau_{by})'$, is now a function of both non-linear velocity gradients and temperature.

## 2.1 Quantifying the bottom stress term

In the equations of motion (14) and (15), we define the bottom stress term using a Herschel-Bulkley model (Herschel and Bulkley, 1926),

$$\boldsymbol{\tau}_b \;=\; \boldsymbol{\tau}_{z\mathbf{x}} \;=\; \mu\frac{\partial \mathbf{u}}{\partial z} \;+\; \tau_{\text{yield}}\left[\text{sgn}\left(\frac{\partial \mathbf{u}}{\partial z}\right)\right], \tag{18}$$

where $\boldsymbol{\tau}_b \;=\; \boldsymbol{\tau}_{z\mathbf{x}} = (\tau_{zx},\, \tau_{zy})'$, $\tau_{\text{yield}}$ is the yield strength of the fluid, sgn denotes the sign of the argument, and $\mu$ is the non-linear viscosity defined as,

$$\mu \;=\; \mathcal{K}\left|\frac{\partial \mathbf{u}}{\partial z}\right|^{n-1}. \tag{19}$$

The symbol $\mathcal{K}$ in equation (19) represents the consistency of the lava and can be modeled solely as a function of temperature while the power law exponent, $n$, is typically a function of the particle content (crystals and/or bubbles) of the lava (which in turn can be a function of temperature), see Castruccio et al. (2010) and Castruccio et al. (2014) for example. We quantify the temperature dependency of the lava consistency ($\mathcal{K}$) in a fashion similar to Sonder et al. (2006) and use the VFT silicate melt model of Giordano et al. (2008),

$$\log\left(\frac{\mathcal{K}}{\mathcal{K}_0}\right) \;=\; A + \frac{B}{T(\text{K}) - C} \tag{20}$$

where $A$ is the value of $\log\mathcal{K}/\mathcal{K}_0$ at infinite temperature $\mathcal{K}_0$ is a constant set to 1 s$^{n-1}$) and $B$ and $C$ are parameters that depend on the composition of the lava. The model assumes

that $A$ is a constant for all silicate melts regardless of composition, and thus, it represents the high temperature limit for silicate melt viscosity. Once the parameter $A$ is fixed then the parameters $B$ and $C$ are determined via a linear ensemble of combinations of oxide components and a subordinate number of multiplicative oxide cross terms, see Giordano et al. (2008) for the full details of the model.

The power law exponent of the viscosity term quantifies the effect that stress gradients have on the material properties of the fluid. A value of $n = 1$ corresponds to a Newtonian fluid while $n < 1$ or $n > 1$ corresponds to a non-Newtonian fluid. If $n > 1$ then the fluid viscosity increases with increasing shear rate (shear thickening) while if $n < 1$ the fluid viscosity decreases with increasing shear rate (known as shear thinning). Typically, if the lava is sufficiently hot and degassed, then the lava stress can be modelled with a Newtonian approximation and $n = 1$. However, if bubbles and/or crystals are present in the lava (depending on the lava source and the amount of degassing that has occurred) then these structures will deform and realign under an applied shear stress. This consequently causes the viscosity of the lava to become thinner in some situations and thicker in others depending on how the structures rearrange. Lava flows with a high crystal content are typically pseudoplastic and shear thinning; the crystal structure of the lava resists the flow of the lava and the lava will not flow unless a yield strength is surpassed. In this case, the lava will continue to flow more readily as the shear stress increases. The opposite tends to occur when a lava flow has a high bubble content at higher capillary numbers; large stress gradients in the flow cause the bubbles to rearrange in a fashion that increases the viscosity and the lava behaves as a shear thickening fluid.

The bottom stress term is a function of the velocity gradient evaluated at the bottom boundary which we do not have access to in the depth-averaged equations, and therefore, we define the bottom stress in terms of the depth-averaged velocity as,

$$\tau_{b\mathbf{x}} \equiv \bar{\mu}\frac{\bar{\mathbf{u}}}{\boldsymbol{\delta_z}} + \tau_{\text{yield}}\left[\text{sgn}\left(\frac{\bar{\mathbf{u}}}{\boldsymbol{\delta_z}}\right)\right] \approx \mu\frac{\partial\mathbf{u}}{\partial z} + \tau_{\text{yield}}\left[\text{sgn}\left(\frac{\partial\mathbf{u}}{\partial z}\right)\right], \qquad (21)$$

where

$$\bar{\mu} = \mathcal{K}\left|\frac{\bar{\mathbf{u}}}{\boldsymbol{\delta_z}}\right|^{n-1}. \qquad (22)$$

It can be noted that in the expression above, $\boldsymbol{\delta_z} = (\delta_{z_x}, \delta_{z_y})'$, is a measure of a *virtual* length over which the shear stress is applied. We determine $\boldsymbol{\delta_z}$ by taking into account the vorticity of the lava flow field, which is defined as,

$$\boldsymbol{\omega} = \left(\frac{\partial w}{\partial y} - \frac{\partial v}{\partial z}\right)\hat{i} + \left(\frac{\partial w}{\partial x} - \frac{\partial u}{\partial z}\right)\hat{j} + \left(\frac{\partial u}{\partial y} - \frac{\partial v}{\partial x}\right)\hat{k}, \qquad (23)$$

where $\hat{i}$, $\hat{j}$, and $\hat{k}$ are unit vectors in the $x-$, $y-$, and $z-$directions, respectively. We solve an auxiliary problem over a *pseudo* depth of the lava that consists of an upper mixed layer where $\mathbf{u}(x, y, z) \equiv \bar{\mathbf{u}}(x, y)$ and a lower layer where $(\partial u/\partial z, \partial v/\partial z)' >> (\partial w/\partial x, \partial w/\partial y)'$. We assume that the vorticity in the upper layer is equal to the vorticity in the bottom layer (in terms of magnitude) at the coordinate point where $\partial\mathbf{u}/\partial z \neq 0$ (i.e., at the interface between the two-layers). This allows us to calculate the vorticity in the upper layer and then use this value to determine $\boldsymbol{\delta_z}$. In other words, we answer the following question;

given a measure of the vorticity associated with the depth averaged velocity field, what is the associated length scale over which the depth-averaged velocity must decay to a value of zero (the bottom boundary condition) to ensure that the internal vorticity of the flow is conserved? (It can be noted that an implicit assumption in depth-integrated models is that internal friction in the vertical is null compared to the friction at flow boundaries. This along with assumption i. and a constant density implies conservation of vorticity about the $\hat{i}$ and $\hat{j}$ directions except at flow boundaries). The key to this approach relies on calculating a measure for the vertical velocity in the upper layer, which we achieve by making use of the kinematic boundary condition,

$$\frac{\partial \zeta}{\partial t} \; + \; u\frac{\partial \zeta}{\partial x}\bigg|_{z=\zeta} \; + \; v\frac{\partial \zeta}{\partial y}\bigg|_{z=\zeta} \; = \; w\bigg|_{z=\zeta}, \tag{24}$$

coupled with the depth-integrated continuity equation (13) to obtain a measure of the vertical velocity $w$. More specifically, expanding derivatives in (13), solving for $\partial \zeta/\partial t$ while substituting this result into (24) and noting that in the upper layer, $(\bar{u}, \bar{v}, \bar{w}) \equiv (u(\zeta), v(\zeta), w(\zeta))$, yields,

$$\bar{w} = \zeta\frac{\partial \bar{u}}{\partial x} \; + \; \zeta\frac{\partial \bar{v}}{\partial y}. \tag{25}$$

The relevant voriticty terms in (23) include the $\hat{i}$ and $\hat{j}$ components. By definition, $\partial \bar{u}/\partial z = \partial \bar{v}/\partial z = 0$ over the upper layer so that the vorticity component about the $x-$axis is $\partial \bar{w}/\partial y$ and the vorticity component about the $y-$axis is $\partial \bar{w}/\partial x$. Because the bottom boundary condition is modelled as a rigid wall where $\mathbf{u} = 0$, and because the fluid is incompressible, a vorticity layer forms in the fluid near the solid boundary that resists the local rotation of the fluid (this is the reason why the rigid boundary does not deform). The vorticity created at the boundary resists the rotation of the interior and is equal to $\partial v/\partial z$ about the $x-$axis and $\partial u/\partial z$ about the $y-$axis (see Schlichting et al. (1968)). Now, if we assume that each vorticity component over the bulk of the flow is equal to each vorticity component in boundary layer at the coordinate point where $(\partial u/\partial z, \partial v/\partial z)'$ is no longer equal to zero, then the virtual length over which the shear stress is applied is given by,

$$\delta_{z_x} \; = \; \frac{\bar{u}}{\partial \bar{w}/\partial x} \quad \text{and} \quad \delta_{z_y} \; = \; \frac{\bar{v}}{\partial \bar{w}/\partial y}. \tag{26}$$

It can be noted that as $(\partial \bar{w}/\partial x, \partial \bar{w}/\partial y)$ goes to 0, the vertical stress in the fluid goes to 0. We can rewrite expression (26) solely in terms of the depth-averaged variables using equation (25),

$$\delta_{z_x} \; = \; \bar{u}\left[\frac{\partial}{\partial x}\left(\zeta\frac{\partial \bar{u}}{\partial x} \; + \; \zeta\frac{\partial \bar{v}}{\partial y}\right)\right]^{-1} \quad \text{and} \quad \delta_{z_y} \; = \; \bar{v}\left[\frac{\partial}{\partial y}\left(\zeta\frac{\partial \bar{u}}{\partial x} \; + \; \zeta\frac{\partial \bar{v}}{\partial y}\right)\right]^{-1}. \tag{27}$$

It can be noted that even though we do not explicitly include horizontal shear stresses in the Kilauea simulations presented in §5 due to the high $Re$ number, the virtual length used to quantify the bottom stress as defined in (27) is a function of horizontal shear within the fluid. Further, we wish to emphasize that the virtual length, $\boldsymbol{\delta}_z$, is non-physical, and is not necessarily less than the lava flow thickness ($H$); it merely is a measure to ensure that $(\bar{\mathbf{u}} - \mathbf{0})/\boldsymbol{\delta}_z \approx \partial \mathbf{u}/\partial z$ at the bottom boundary in a fashion that conserves internal vorticity,

i.e., it ensures that the interior of the flow field is irrotational about the $\hat{i}$ and $\hat{j}$ coordinate axis.

## 2.2 Heat transfer

As soon as lava effuses from an active vent it begins to degas and transfer heat to its surroundings. Lava cools through the mechanisms of radiation, conduction and convection in the air above it (we neglect heating from viscous dissipation, which is small compared to heat loss through radiation and conduction for the low-viscosity flows we are considering here (e.g., Harris and Rowland, 2001)). We quantify heat loss due to radiation via Stefan's law (Griffiths, 2000),

$$q_s \;=\; \frac{\epsilon \sigma_B}{\rho c_p} \left( \bar{T}^4 - T_{\text{atm}}^4 \right), \tag{28}$$

where $\epsilon$ is the emissivity of the lava, $\sigma_B$ is the Stefan-Boltzmann constant, and $T_{\text{atm}}$ is the temperature of the surrounding atmosphere in degree Kelvin. When lava temperatures fall below the solidus (e.g., $\sim 950\ C$ for Kilauea lavas), buoyancy driven convection in the air above the lava becomes the dominant mode of heat transfer at the lava surface instead of radiation (due to crust formation) (Griffiths, 2000). In this case we set $q_s$ in the energy equation to,

$$q_s \;=\; k_c \left( \bar{T} - T_{\text{atm}} \right)^{4/3}, \tag{29}$$

where $k_c$ is a heat transfer coefficient (e.g., Patrick et al., 2004) for more details. We quantify heat transfer from the lava to the ground via conduction; in symbols we have (Patrick et al., 2004),

$$q_b \;=\; k_b \left( \bar{T} - T_{\text{ground}} \right), \tag{30}$$

where $T_{\text{ground}} = f(\mathbf{x})$ is the temperature of the ground in contact with the lava flow field and $k_b$ measures the thermal conductivity of the ground. We utilize equation (30) to determine the temperature near the bottom boundary of lava flow field which we use to evaluate the nonlinear viscosity in the bottom stress term in the equations of motion. More specifically, we can re-write (30) in terms of a depth-dependent thermal boundary-layer temperature, $T(z)$,

$$\frac{\partial T}{\partial z} \;=\; \frac{\tilde{k}_b}{h_b} \left( T - T_{\text{ground}} \right), \tag{31}$$

where $\tilde{k}_b$ is the thermal boundary layer conductivity constant and $h_b(x,y)$ is the thickness of the thermal boundary layer (see Figure 2). We solve equation (31) over the thermal boundary layer defined in the $z-$direction from $z = z_b(x,y)$ to $z = -h(x,y)$ by setting $z_b(x,y)$ to a relative zero. We then integrate equation (31) over a thermal boundary coordinate defined from $\tilde{z} = 0$ to $\tilde{z} = -h_b(x,y)$. (It can be noted that the relationship between $\tilde{z}$ and $z$ is given

by $\tilde{z} = z - z_b$ so that $h_b = h - z_b$). Equation (31) is a non-homogeneous, constant coefficient ordinary differential equation that has the solution,

$$T(\tilde{z}) \;=\; (T_{\text{int}} - T_{\text{ground}}) \exp\left(\frac{\tilde{k}_b}{h_b}\tilde{z}\right) \;+\; T_{\text{ground}}, \tag{32}$$

which gives an expression for the temperature profile over the thermal boundary layer of the lava ($\tilde{z} \in [0, -h_b]$). Evaluating equation (32) at $\tilde{z} = -h_b$ and setting the interior temperature ($T_{\text{int}}$) to the depth-integrated value ($\bar{T}$), we have,

$$T\bigg|_{\tilde{z}=-h_b} \;=\; (\bar{T} - T_{\text{ground}}) \exp\left(-\tilde{k}_b\right) \;+\; T_{tnground}. \tag{33}$$

We use this temperature to evaluate the consistency in the bottom stress approximation,

$$\mathcal{K}\bigg|_{z=-h} = \mathcal{K}\left(T\bigg|_{\tilde{z}=-h_b}\right). \tag{34}$$

The greater the thermal conductivity of the boundary layer, the closer the boundary temperature is to the ground temperature, however, in general, there is usually a steep gradient in the temperature at the interface between the boundary of the flowing lava and the ground that the lava is conducting heat to. It can be noted that we also use an analogous approach to calculate the temperature of the lava at channel wall boundaries.

## 2.3   Steady reference depth of flow $h$

We have two options to calculate the steady reference depth of flow ($h$) of the lava that we use as a zero datum to measure the free surface from. Our particular choice depends on the inflow data available to the model. For instance, if a full set of temporally varying inflow data is available, we set $h$ equal to the time average thickness associated with the data, i.e.,

$$h = \frac{1}{(t_f - t_i)} \int_{t_i}^{t_f} \frac{\mathbf{Q}_{\text{in}} \cdot \hat{\mathbf{n}}}{\text{w}_{\text{in}}\left(\mathbf{u}_{\text{in}} \cdot \hat{\mathbf{n}}\right)} \, \mathrm{d}t, \tag{35}$$

where $\mathbf{Q}_{\text{in}} \cdot \hat{\mathbf{n}}$ is the inflow flux normal to the boundary and $\text{w}_{\text{in}}$ is the width of the inflow boundary normal to the flow. If, however, the only inflow data available to the model is a set of time-averaged data, then we set $h$ to the solution of the steady, linear system of equations associated with the full nonlinear system of equations given by (17).

# 3   Numerical discretization

To develop our numerical methods, we rewrite the system of equations (1) in the compact form,

$$\frac{\partial U^{(i)}}{\partial t} \;+\; \nabla \cdot \mathbf{F}^{(i)}(\mathbf{U}) \;=\; S^{(i)}(\mathbf{U}), \quad i = 1, \, 2, \, 3, \, 4 \tag{36}$$

where $U^{(i)}$, $\mathbf{F}^{(i)}$, and $S^{(i)}$ are the $i$-th row entries of the vectors $\mathbf{U}$, $\mathbf{S}$, and the flux function matrix $\mathbf{F}$, defined as,

$$
\mathbf{U} = \begin{bmatrix} H \\[6pt] H\bar{u} \\[6pt] H\bar{v} \\[6pt] H\bar{T} \end{bmatrix}, \quad
\mathbf{F} = \begin{bmatrix} H\bar{u}, & H\bar{v} \\[6pt] H\bar{u}^2 + P, & H\bar{u}\bar{v} \\[6pt] H\bar{u}\bar{v}, & H\bar{v}^2 + P \\[6pt] H\bar{u}\bar{T}, & H\bar{v}\bar{T} \end{bmatrix}, \quad
\mathbf{S} = \begin{bmatrix} 0 \\[6pt] -g_x H + g\zeta \dfrac{\partial h}{\partial x} - \dfrac{\tau_{bx}}{\rho} \\[6pt] -g_y H + g\zeta \dfrac{\partial h}{\partial y} - \dfrac{\tau_{by}}{\rho} \\[6pt] q_s + q_b \end{bmatrix},
$$

where $P = \frac{1}{2} g_z \left( H^2 - h^2 \right)$.

## 3.1 Finite element partition

To apply a DG spatial discretization to our mathematical model (36) over a lava flow channel (see Figure 4 for example), we begin by introducing a partition of the two-dimensional domain $\Omega$. The complexities of the domain boundary, $\partial\Omega$, are such that an unstructured finite element partition (or mesh) is necessary to properly capture its intricacies. More specifically, we obtain unstructured triangulations (that we denote by $\mathcal{T}_h$) of the channel domain $\Omega$ via an automatic mesh generator known as ADMESH$^+$ (Conroy et al., 2012). ADMESH$^+$ solves a number of differential equations to calculate a mesh size function that determines local element sizes based on the curvature of the boundary, channel width, and changes in the topography and domain slope to create a high-quality unstructured simplex mesh (the elements are close to equilateral triangles). The only input required by the program is a list of points defining the boundary as well as the topography of the domain.

## 3.2 A weak form and the semi-discrete equations

Given the finite element partition, $\mathcal{T}_h$, of the domain $\Omega$, we obtain a weak form of equation (36) if we first multiply (36) by a sufficiently smooth test function $\psi(x, y) \in \mathcal{V}$, integrate over each element $\Omega_j \in \mathcal{T}_h$, and then integrate the flux term by parts,

$$
\int_{\Omega_j} \frac{\partial U^{(i)}}{\partial t} \, \psi \, dA - \int_{\Omega_j} \mathbf{F}^{(i)} \cdot \nabla \psi \, dA + \int_{\partial\Omega_j} \left( \mathbf{F}^{(i)} \cdot \hat{\mathbf{n}} \right) \psi \, dS = \int_{\Omega_j} S^{(i)} \, \psi \, dA, \qquad U^{(i)}, \psi \in \mathcal{V},
\tag{37}
$$

for $i = 1$, 2, 3, 4, and $j = 1, \ldots, \mathcal{N}$, where $\mathcal{N}$ is the total number of elements of the triangulation $\mathcal{T}_h$. In the equation above, $\hat{\mathbf{n}}$ is the outward unit normal to the element boundary $\partial\Omega_j$. Rather than seek solutions to (37) we search for solutions in the finite dimensional subspace of functions defined as

$$
\mathcal{V}_{hp} = \left\{ \psi : \psi \big|_{\Omega_j} \in \mathcal{P}_\ell \left( \Omega_j \right), \forall \Omega_j \right\},
\tag{38}
$$

where $\mathcal{P}_\ell$ demarcates the space of polynomials of at most degree $\ell$ that is not necessarily continuous across element boundaries. In other words, given a set of basis functions $\phi = (\phi_0, \phi_1, \ldots, \phi_\ell)'$, we express the trial solution ($U_h^{(i)} \in \mathcal{V}_{hp}$) and test function ($\psi_h \in \mathcal{V}_{hp}$) as

$$U_h^{(i)}\bigg|_{\Omega_j} = \sum_{l=0}^{\ell} U_l^{(i)}(t)\phi_l(\mathbf{x}), \tag{39}$$

and

$$\psi_h\bigg|_{\Omega_j} = \sum_{l=0}^{\ell} \psi_l(t)\phi_l(\mathbf{x}), \tag{40}$$

where $\left(U_0^{(i)}, U_1^{(i)}, \ldots, U_\ell^{(i)}\right)'$ are the time-dependent degrees of freedom of the finite element solution and $i = 1, 2, 3, 4$. We use products of Jacobi polynomials of degree $\ell$, $\{\mathcal{P}_l\}_{l=0}^{\ell}$, as the basis for $\mathcal{V}_{hp}$. The orthogonal triangular basis is defined in terms of a "collapsed coordinate" system that results in a matrix free implementation of the method, see Kubatko et al. (2006) for more details. Substituting $U_h^{(i)}$ and $\psi_h$ into (37) we arrive at the discrete weak form of the problem: find $U_h^{(i)} \in \mathcal{V}_{hp}$ such that for all test functions $\psi_h \in \mathcal{V}_{hp}$, for $i = 1, 2, 3, 4$, the expression,

$$\int_{\Omega_j} \frac{\partial U_h^{(i)}}{\partial t} \, \psi_h \, dA \; - \; \int_{\Omega_j} \mathbf{F}^{(i)}(\mathbf{U}_h) \cdot \nabla \psi_h \, dA \; + \; \int_{\partial\Omega_j} \left(\hat{\mathbf{F}}^{(i)} \cdot \hat{\mathbf{n}}\right) \psi_h \, dS \; = \; \int_{\Omega_j} S^{(i)}(\mathbf{U}_h) \, \psi_h \, dA, \tag{41}$$

holds over each element $\Omega_j \in \mathcal{T}_h$, where $S^{(i)}(\mathbf{U}_h)$ is the source term evaluated in $\mathcal{V}_{hp}$ and $\hat{\mathbf{F}}^{(i)}$ is a suitably chosen numerical flux.

### 3.2.1 Numerical flux

The space of functions defined by (38) is not necessarily continuous across element boundaries, and thus, can be dual-valued (see Figure 3 for example). To remedy this inconsistency, we replace the dual-valued flux in (37) with a so-called numerical flux ($\hat{\mathbf{F}}$) that makes use of the left and right limits of the trial solution to produce a single valued flux across a given element's boundary.

More specifically, given an arbitrary function $w_h \in \mathcal{V}_{hp}$ at an element boundary point $\mathbf{x}_i$, we set the left and right limits of the function to $w_h^- \equiv w_h(\mathbf{x}_i^-)$ and $w_h^+ \equiv w_h(\mathbf{x}_i^+)$, respectively. In this work we utilize the local Lax-Friedrichs (LLF) flux, which defines the numerical flux operator as,

$$\hat{\mathbf{F}}^{(i)} \cdot \hat{\mathbf{n}} \; = \; \frac{1}{2}\left(\mathbf{F}^{(i,+)} + \mathbf{F}^{(i,-)}\right) \cdot \hat{\mathbf{n}} \; - \; \frac{1}{2}|\lambda_{\max}|\left(U_h^{(i,+)} - U_h^{(i,-)}\right), \quad \text{for} \;\; i = 1, 2, 3, 4, \tag{42}$$

where $\lambda_{\max}$ is the maximum eigenvalue of the normal (to the element edges) Jacobian matrix.

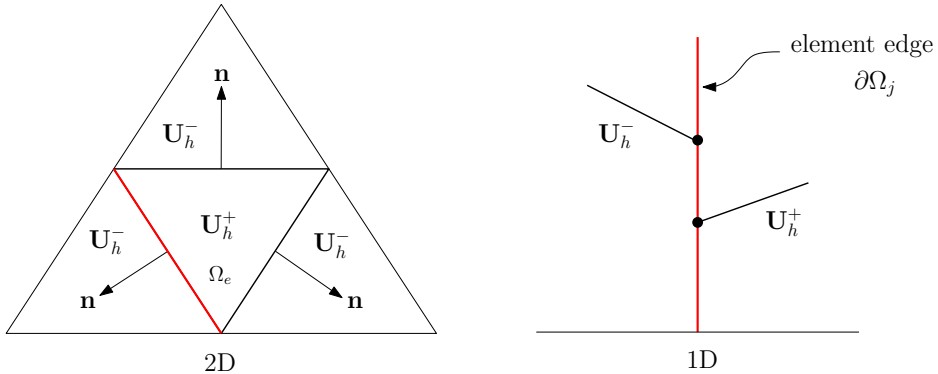

Figure 3: Jump in numerical solution $\mathbf{U}_h$ at an element edge $\partial\Omega_j$.

When solutions to (36) are sufficiently smooth, we can rewrite (36) in the quasilinear form,

$$\frac{\partial \mathbf{U}}{\partial t} + \mathbf{J}_x(\mathbf{U})_x + \mathbf{J}_y(\mathbf{U})_y = \mathbf{S}, \tag{43}$$

where the Jacobian matrices $(J_{ij} = \frac{\partial F_i}{\partial x_j})$ are,

$$\mathbf{J}_x = \begin{bmatrix} 0 & 1 & 0 & 0 \\ g_z H - \bar{u}^2 & 2\bar{u} & 0 & 0 \\ -\bar{u}\bar{v} & \bar{v} & \bar{u} & 0 \\ -\bar{u}\bar{T} & \bar{T} & 0 & \bar{u} \end{bmatrix},$$

and

$$\mathbf{J}_y = \begin{bmatrix} 0 & 0 & 1 & 0 \\ -\bar{u}\bar{v} & \bar{v} & \bar{u} & 0 \\ g_z H - \bar{v}^2 & 0 & 2\bar{v} & 0 \\ -\bar{v}\bar{T} & 0 & \bar{T} & \bar{v} \end{bmatrix}.$$

406     The so-called "normal Jacobian matrix" is then defined by,

$$\mathbf{J}_n = \mathbf{J}_x n_x + \mathbf{J}_y n_y, \tag{44}$$

where $n_x$ and $n_y$ are the $x-$ and $y-$components of the normal edge vector $\hat{\mathbf{n}}$. In general, if $\mathbf{J}$ is a square $(m \times m)$ matrix with $m$ real eigenvalues, then it can be decomposed into its eigensystem,

$$\mathbf{J}_x = \mathcal{R}_{(x)} \mathbf{\Lambda}_{(x)} \mathcal{R}_{(x)}^{-1}, \quad \text{and} \quad \mathbf{J}_y = \mathcal{R}_{(y)} \mathbf{\Lambda}_{(y)} \mathcal{R}_{(y)}^{-1}, \tag{45}$$

where $\mathcal{R}_{(\cdot)}$ is the matrix of right eigenvectors, $\mathbf{\Lambda}_{(\cdot)}$ is the diagonal matrix of eigenvalues, and $\mathcal{R}_{(\cdot)}^{-1}$ is the matrix of left eigenvectors (LeVeque, 2002). To determine $\Lambda_{(x)}$ and $\Lambda_{(y)}$ we solve for the roots of $\det\left(\mathbf{J}_{(\cdot)} - \lambda\mathcal{I}\right) = 0$, which gives the following eigenvalues,

$$
\begin{aligned}
\lambda_{1,2} &= \bar{u}n_x + \bar{v}n_y, \\
\lambda_3 &= \left(\bar{u} + \sqrt{g_z H}\right)n_x + \left(\bar{v} + \sqrt{g_z H}\right)n_y, \\
\lambda_4 &= \left(\bar{u} - \sqrt{g_z H}\right)n_x + \left(\bar{v} - \sqrt{g_z H}\right)n_y.
\end{aligned}
\tag{46}
$$

Each eigenvector $(\mathbf{r}_i^{(\cdot)})$ can be determined by solving $(\mathbf{J}_{(\cdot)} - \lambda_i\mathbf{I})\mathbf{r}_i = \mathbf{0}$ where $\mathbf{I}$ is the identity matrix, $\mathbf{0}$ is a vector of zeros, and $\mathcal{R}_{(\cdot)} = \left[\mathbf{r}_1^{(\cdot)}, \mathbf{r}_2^{(\cdot)}, \mathbf{r}_3^{(\cdot)}, \mathbf{r}_4^{(\cdot)}\right]$. Solving for the eigenvectors we have,

$$
\mathcal{R}_{(x)} = \begin{bmatrix}
0 & 0 & 1/\bar{T} & 1/\bar{T} \\
0 & 0 & (\bar{u} + \sqrt{g_z H})/\bar{T} & (\bar{u} - \sqrt{g_z H})/\bar{T} \\
1 & 0 & \bar{v}/\bar{T} & \bar{v}/\bar{T} \\
0 & 1 & 1 & 1
\end{bmatrix},
$$

and

$$
\mathcal{R}_{(y)} = \begin{bmatrix}
0 & 0 & 1/\bar{T} & 1/\bar{T} \\
1 & 0 & \bar{u}/\bar{T} & \bar{u}/\bar{T} \\
0 & 0 & (\bar{v} + \sqrt{g_z H})/\bar{T} & (\bar{v} - \sqrt{g_z H})/\bar{T} \\
0 & 1 & 1 & 1
\end{bmatrix}.
$$

We use the full eigensystem in the slope limiting process that stabilizes the method for polynomials of degree greater than or equal to one (Cockburn and Shu, 2001), and we set $\lambda_{\max}$ in the LLF flux to the maximum value of $(\lambda_1, \lambda_2, \lambda_3, \lambda_4)$.

It can be noted that to mathematically close the solution method of the discrete DG system of equations we need to numerically evaluate expression (27) to determine $\boldsymbol{\delta}_z$. More specifically, we discretize equation (27) using a local discontinuous Galerkin (LDG) method (Cockburn and Chi-Wang, 1998) analogous to the method used in Conroy and Kubatko (2016) to evaluate second order derivative terms (see Conroy (2014) for a detailed discussion on application of the LDG method to shallow mass geophysical fluid flows).

## 3.3   SSP Runge-Kutta time discretizations

Application of the DG spatial operator to (41) results in a system of ODEs for each element,

$$\tilde{\mathbf{M}}_j^{(i)} \frac{d\tilde{\mathbf{U}}_j^{(i)}}{dt} = \mathbf{b}_j^{(i)}, \quad i = 1,\ 2,\ 3,\ 4 \quad \text{and} \quad j = 1, ..., \mathcal{N} \tag{47}$$

where $\mathcal{N}$ is the number of elements in $\mathcal{T}_h$, $\tilde{\mathbf{U}}_j^{(i)} = \left( U_{j,0}^{(i)}, U_{j,1}^{(i)}, \ldots, U_{j,\ell}^{(i)} \right)'$ are vectors of the degrees of freedom (i.e., the polynomial basis coefficients), and $\mathbf{b}_j^{(i)} = \left( \mathbf{R}_{j,0}^{(i)}, \mathbf{R}_{j,1}^{(i)}, \ldots, \mathbf{R}_{j,\ell}^{(i)} \right)'$ with,

$$\mathbf{R}_{j,l}^{(i)} = \int_{\Omega_j} \mathbf{F}_j^{(i)} \cdot \nabla\phi_l \ dA - \int_{\partial\Omega_j} \left( \hat{\mathbf{F}}_j^{(i)} \cdot \mathbf{n} \right) \phi_l \ dS + \int_{\Omega_j} S_j^{(i)} \ \phi_l \ dA. \tag{48}$$

$\tilde{\mathbf{M}}_j^{(i)}$ is the mass matrix,

$$\tilde{\mathbf{M}}_j^{(i)} = \begin{bmatrix} \int_{\Omega_j} \phi_1\phi_1 \ dA & 0 & \ldots & & 0 \\ 0 & \int_{\Omega_j} \phi_2\phi_2 \ dA & 0 & & \vdots \\ \vdots & 0 & \ddots & 0 & \\ 0 & \ldots & & 0 & \int_{\Omega_j} \phi_\ell\phi_\ell \ dA \end{bmatrix}.$$

which is diagonal due to the choice of basis. Left multiplying (47) by the inverse of the mass matrix, we have,

$$\frac{d\tilde{\mathbf{U}}^{(i)}}{dt} = \left( \tilde{\mathbf{M}}_j^{(i)} \right)^{-1} \mathbf{b}_j^{(i)} = \mathcal{L}_{hp}(\tilde{\mathbf{U}}), \quad \text{with} \ \ i = 1,\ 2,\ 3,\ 4, \ \text{and} \ \ j = 1, ..., \mathcal{N}, \tag{49}$$

where $\mathcal{L}_{hp}$ is the DG spatial operator. We evaluate the integrals in equation (48) using numerical integration rules of sufficiently high degree (Kubatko et al., 2006), and discretize (49) with so-called strong-stability-preserving (SSP) Runge–Kutta (RK) methods (Kubatko et al., 2014). The unknown polynomial basis coefficients that define the solution over a given element, $\Omega_j$, are advanced in time from $t_m$ to $t_{m+1}$ via,

1. Set $\tilde{\mathbf{U}}_0^{(i)} \leftarrow \tilde{\mathbf{U}}_m^{(i)}$, for $i = 1,\ 2,\ 3,\ 4$.

2. For each stage $r = 1, 2, \ldots, \mathcal{S}$, set

$$\tilde{\mathbf{U}}_r^{(i)} \leftarrow \Pi_h \left( \sum_{s=1}^{r} \alpha_{rs} \mathbf{w}^{rs} \right), \quad \mathbf{w}^{rs} = \tilde{\mathbf{U}}_{s-1}^{(i)} + \frac{\beta_{rs}}{\alpha_{rs}} \Delta t \ \mathbf{L}_h \left( \tilde{\mathbf{U}}_{s-1}, t_m + \delta_s \Delta t \right).$$

3. Finally, set $\tilde{\mathbf{U}}_{m+1}^{(i)} \leftarrow \tilde{\mathbf{U}}_{\mathcal{S}}^{(i)}$.

It can be noted that $\Pi_h$ is a slope limiter that dampens over shoots and under shoots at solution discontinuities when polynomial approximations greater than 0 are used for the basis (Cockburn and Shu, 2001), $\mathcal{S}$ is the number of stages of the RK method, $\delta_s \Delta t$ is a sub-time step of the time step $\Delta t$, and the $\alpha_{rs}$ and $\beta_{rs}$ are coefficients that define the RK method. In particular, $\alpha_{rs}$ and $\beta_{rs}$ conform to the following constraints,

1. $\alpha_{rs} = 0$ if and only if $\beta_{rs} = 0$,

2. $\alpha_{rs} \geq 0$ and $\beta_{rs} \geq 0$,

3. $\sum_{s=1}^{r} \alpha_{rs} = 1$.

Because we use explicit RK methods the time step of the model is limited by a CFL condition, see Kubatko et al. (2014) for more details.

# 4    Verification

Verification of the DG solution of the mass and momentum equations in the depth-averaged and full three-dimensional case is well documented and can be found in (Conroy and Kubatko, 2016), (Dawson and Aizinger, 2005) and (Kubatko et al., 2006). To verify our DG solution method for the fully coupled mass, momentum, and energy (depth-averaged) equations we solve a test problem that is designed to model a free surface wave propagating through a lava channel using the method of manufactured solutions (see Griffiths (2000) for a discussion on free surface waves in lava channels at high $Re$ number and Le Moigne et al. (2020) for a detailed investigation on standing waves in lava flow channels). We choose $\bar{u}$, $\bar{v}$, and $H$ so that the depth-averaged mass equation is satisfied exactly. Specifically, we define,

$$\bar{u} = \hat{u}\exp(-x),$$

$$\bar{v} = \hat{v}\exp(-kx),$$

$$H = h + \hat{\zeta}\exp(\mathrm{i}\,\omega t), \tag{50}$$

with $h = \text{constant}$, $\hat{u} = \text{constant}$, $\hat{v} = \hat{u}ky$, and $\hat{\zeta} = \exp(-\mathrm{i}\omega\hat{x})$ where $\hat{x} = \exp(kx)/(k\hat{u})$. The dynamical solution consists of a wave propagating in a direction perpendicular to the y-axis with wave number $k = 5.0 \times 10^{-3}$ m$^{-1}$ and frequency $\omega = 2.0 \times 10^{-3}$ s$^{-1}$. (These values were chosen based on velocity and lava flow thickness data recorded by Patrick et al. (2019) during the pulsing effusion regime associated with Fissure 8 during the 2018 Kilauea event.) We then set,

$$\bar{T} = \hat{T}\exp(-k_T x), \tag{51}$$

with $\hat{T} = (y^2 + T_{\text{wall}})$ and substitute (50) and (51) into the mathematical model (17) and evaluate the derivative terms using Matlab's symbolic package. The remainder terms associ-

Table 1: $L^2$ errors using $\mathcal{P}_0$ for $(\zeta_h, \bar{u}_h, \bar{v}_h, \bar{T}_h)$.

| Mesh | $||\zeta_h - \zeta||_2$ | Order | $||\bar{u}_h - \bar{u}||_2$ | Order | $||\bar{v}_h - \bar{v}||_2$ | Order | $||\bar{T}_h - \bar{T}||_2$ | Order |
|---|---|---|---|---|---|---|---|---|
| $dx_0$ | 3.91 | – | 8.81 | – | 9.24 | – | 2126.3 | – |
| $dx_1$ | 2.06 | 0.92 | 4.48 | 0.98 | 4.82 | 0.94 | 1227.7 | 0.79 |
| $dx_2$ | 1.39 | 0.97 | 3.01 | 0.98 | 3.30 | 0.93 | 875.11 | 0.84 |
| $dx_3$ | 0.31 | 1.08 | 0.77 | 0.98 | 0.77 | 1.05 | 218.2 | 1.00 |

Table 2: $L^2$ errors using $\mathcal{P}_1$ for $(\zeta_h, \bar{u}_h, \bar{v}_h, \bar{T}_h)$.

| Mesh | $||\zeta_h - \zeta||_2$ | Order | $||\bar{u}_h - \bar{u}||_2$ | Order | $||\bar{v}_h - \bar{v}||_2$ | Order | $||\bar{T}_h - \bar{T}||_2$ | Order |
|---|---|---|---|---|---|---|---|---|
| $dx_0$ | 2.30e-2 | – | 0.33 | – | 5.82e-2 | – | 1589.8 | – |
| $dx_1$ | 4.85e-3 | 2.25 | 6.5e-2 | 2.35 | 1.50e-2 | 1.95 | 367.7 | 2.11 |
| $dx_2$ | 1.72e-3 | 2.56 | 2.09e-2 | 2.79 | 6.33e-3 | 2.13 | 154.3 | 2.14 |
| $dx_3$ | 1.84e-4 | 1.63 | 2.13e-3 | 1.65 | 4.32e-4 | 1.94 | 6.73 | 2.26 |

ated with the x-momentum equation and the energy equation are then set as artificial source terms that force the numerical solution to be (50) and (51). The numerical domain consists of a rectangular channel defined by the Cartesian-coordinates $x_0 = 0.0$m, $x_L = 200.0$m, $y_0 = 0.0$m, $y_L = 30.0$m. We assume symmetry about the centerline (at $y = 15$m) and only solve the equations over the half-width of the channel. It can be noted that even though the solutions are guaranteed to remain smooth for all time $t$ (because of the forcing functions) the numerical solution is by no means trivial due to the coupling of the equations through the viscosity. We use four different triangular meshes for the verification of the model. The element size of each mesh is 7.50 m (the so-called $dx_0$ mesh), 3.75 m ($dx_1$), 2.50 m ($dx_2$), and 0.625 m ($dx_3$), respectfully. Results are displayed in Tables 1-3 where it can be noted that the method converges to the analytic solution at a rate of approximately $\ell + 1/2$. Further, using $\mathcal{P}_2$ polynomials on the coarsest mesh gives lower errors than $\mathcal{P}_0$ polynomials on the finest mesh. It can be noted that for a given computational mesh, a higher order polynomial approximation will result in a greater computational expense. However, the goal of using high order polynomial approximations is to use coarser meshes which results in better computational efficiency in terms of the number of degrees of freedom necessary to achieve a certain level of accuracy (high order local polynomials produce more accurate results more efficiently than low order methods). This is explicitly shown in the works of Kubatko et al. (2006), Kubatko et al. (2009), and Conroy et al. (2018).

# 5  Evaluation: Recent eruption of Kilauea Volcano

We evaluate our model using data captured during the 2018 eruption of Kilauea volcano, Hawai'i. The East Rift Zone of Kilauea has erupted repeatedly in historical times, and continuously since 1983 (Heliker and Mattox, 2003; Wolfe, 1988). A new eruption of unusually large magnitude began May 3, 2018 in the lower part of the East Rift Zone, with fissures opening in the middle of a residential area (Neal et al., 2019). More than 20 fissures opened

Table 3: $L^2$ errors using $\mathcal{P}_2$ for $(\zeta_h, \bar{u}_h, \bar{v}_h, \bar{T}_h)$.

| Mesh | $||\zeta_h - \zeta||_2$ | Order | $||\bar{u}_h - \bar{u}||_2$ | Order | $||\bar{v}_h - \bar{v}||_2$ | Order | $||\bar{T}_h - \bar{T}||_2$ | Order |
|------|------|------|------|------|------|------|------|------|
| $dx_0$ | 1.41e-2 | – | 6.76e-2 | – | 6.85e-3 | – | 17.20 | – |
| $dx_1$ | 2.56e-3 | 2.46 | 2.37e-2 | 1.51 | 3.51e-3 | 0.96 | 3.04 | 2.50 |
| $dx_2$ | 8.00e-4 | 2.87 | 8.70e-3 | 2.47 | 1.11e-3 | 2.84 | 1.09 | 2.52 |
| $dx_3$ | 1.24e-4 | 2.68 | 1.46e-3 | 2.57 | 1.70e-4 | 2.70 | .176 | 2.63 |

during the first 12 days of the eruption, erupting slow-moving, unusually high viscosity lava at low effusion rates. The behavior changed on May 18, when much hotter and less viscous lava reached the surface. Advance rates and flow lengths increased, widely impacting property and infrastructure. Complete evacuation orders followed within days. Starting on May 28th, activity focused at Fissure 8, located in the heart of the Leilani Estate subdivision. Fissure 8 remained the source of lava for the remainder of the eruption, until its abrupt stop on August 4th. The lava that erupted from Fissure 8 soon established a channel which flowed north and east of the vent, forming a moderately branched channel network 4 km from the vent. The flow field exhibited transitions between flow types; a clear transition from pahoehoe to 'a'a surface texture occurred down slope and is apparent on the thermal map (see Figure 1). Overall, the Fissure 8 lava covered an area of 25 km$^2$ and supplied at least 1 cubic km of lava (out of at least 1.2 total) over 70 days.

## 5.1 Observational data

During the 2018 Kilauea eruption, Unoccupied Aerial Systems (UAS) captured a comprehensive time-series of overhead videos of channelized lava (the "Fissure 8" flow). The videography campaign was purposefully designed to collect data for 'remote rheometry' by hovering above specific sites spaced 200-1300 m apart along the length of the open channel and revisiting them throughout the duration of the eruption. The proximal (near vent) sites record pahoehoe lava with little crust cover, while the distal sites capture behavior entirely in the 'a'a flow regime. Sites within the braided section of the flow recorded video over parallel channels. Over 500 hover videos at the channel sites were acquired over the course of the Fissure 8 eruption between May 30 and August 5. In this paper, we focus on videos collected at UAS site 8, capturing a junction point where the main channel split into two branches.

### 5.1.1 Velocity field measurements

We analyze the UAS hover videos using the Optical Flow technique (Horn and Schunck, 1981; Sun et al., 2010) Optical Flow is a well-known Computer Vision technique used to measure velocities of imaged objects based on the motion of brightness within an image sequence or between frames of a video. Lev et al. (2012) used Optical Flow to measure the two-dimensional surface velocities of laboratory-scale basaltic lava flows. We will follow the same technique as in Lev et al. (2012), tuning parameters to the specifics of the Kilauea 2018 UAS footage. Length scale for video analysis and channel geometry data are provided from

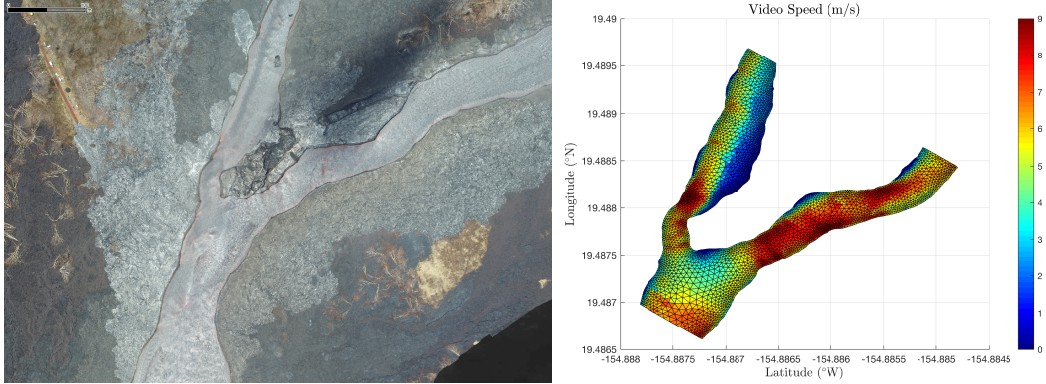

Figure 4: A) Zoomed in view of the section of the braided channel system established by Fissure 8 modeled in this work, near UAS site 8 (See Fig. 1). UAS photo by Ryan Perroy, University of Hawaii-Hilo. B) Map view of the lava surface velocity measured using Optical Flow from videos captured by UAS on June 22nd, 2018. Colors represent magnitude in m/s. Also shown in the discretize finite-element mesh used to evaluate the model.

camera lens information and the recorded UAS flight altitude and refined using co-registered digital elevation and orthomosaic images produced from additional UAS data collected at the same or very close time.

## 5.2  Model input

We provide our model with a channel geometry, assumed material properties, inlet ve-
locity, and observed temperature. We use topography data from a pre-eruption digital elevation maps (data from the USGS National Elevation Dataset, with a spatial resolution of 10m/pixel, USGS (2002)) to calculate the gradient of topography (see Figure 5B). We set the inlet velocity equal to values measured from the UAS video analyzed by Optical Flow (Figure 4B). Channel edge geometry is obtained from the velocity field ($||\mathbf{u}|| >= 0$) com-
bined with visible identification of channel boundaries in the UAS image. Figure 5 shows the meshed model domain, with colors depicting (a) the elevation and (b) ground slope used to set up the model.

We set the the lava density to $\rho = 1350$ kg/m$^3$, which, with a nominal gas-free density of Hawaiian basalts of 2700 kg/m$^3$ translates to 50% vesicularity. We set the channel inlet temperature to $\bar{T} = 1152$ C and wall and basal temperatures to $T_{\text{wall}} = 1010$ C and $T_{\text{ground}} = 477$ C, respectively. The rheological constants in relation (20) are set to $A = -4.550$, $B = 5805.30$ and $C = 607.80$. These values were calculated using the calculator by Giordano et al. (2008) and are specific for the composition of the basalt that erupted during June 2018 from Fissure 8 as measured by XRF analysis (Gansecki et al., 2019)). See Table 4 for the coefficient values used in the heat transfer module. It can be noted that because the lava temperature never falls below 950 C in the Kilauea simulations, surface heat loss is solely due to radiation.

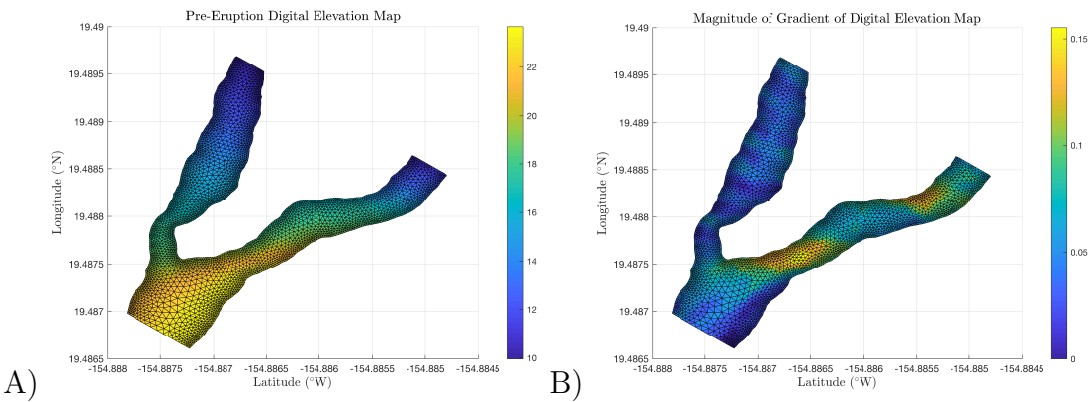

Figure 5: Finite element partition of the modeled section of the braided channel system. Colors corresponds to: (A) topography elevation in meters, and B) ground slope in degrees

Table 4: Value of thermal coefficients.

| Coefficient | Meaning | Value used in Kilauea simulation |
|---|---|---|
| $\epsilon$ | Emissivity | 0.85 |
| $\sigma_b$ | Stefan-Boltzmann constant | $5.670 \times 10^{-8}$ J· m$^{-2}$· s$^{-1}$· K$^{-4}$ |
| $c_p$ | Heat capacity | 837 J · kg$^{-1}$· K$^{-1}$ |
| $k_c$ | heat transfer coefficient | 200.0 W·m$^{-1}$· K$^{-1}$ |
| $k_b$ | thermal conductivity of the ground | 0.90 W·m$^{-1}$· K$^{-1}$ |
| $\tilde{k}_b$ | thermal conductivity of the thermal boundary layer | 0.10 W·m$^{-1}$· K$^{-1}$ |

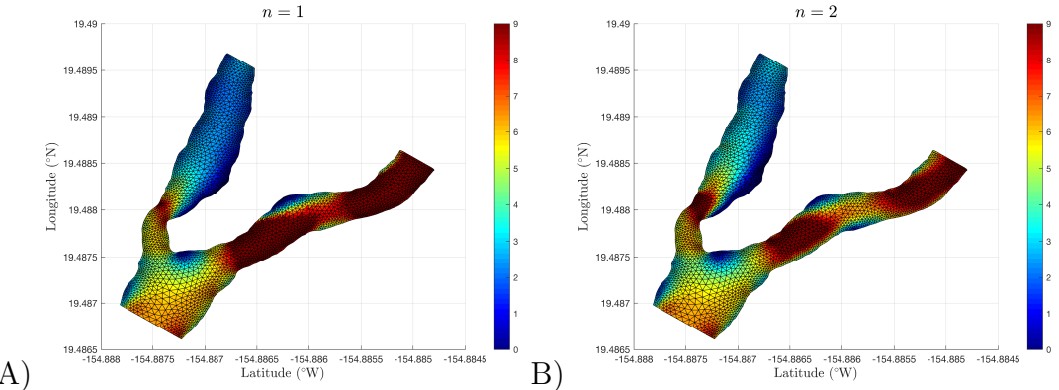

Figure 6: Map view of modeled speed using a value of (A) $n = 1$ and (B) $n = 2$ in the viscosity model. Colors represent velocity magnitude in m/s.

## 5.3 Model results

All model simulations were executed on a Macbook Pro using Intel's Fortran compiler and on average took 56 minutes to execute using a time step dt = 0.05 secs. The finite element
partition of the braided channel system (shown in Figure 5) consists of 6908 elements with a maximum element size of 8 m and a minimum element size of 1 m. The model's initial conditions were set to the solution of a linearized form of (17) with $\zeta = 0$ and $h = 10$ m. Inlet conditions are steady in time and we set $\boldsymbol{\tau}_{\mathrm{yield}} = 0$ in the Herchel-Bulkley model. We set the initial conditions to the solution of the linear equations of (17) and time step the
non-linear system of equations (17) to steady state.

Figure 6 shows the lava velocities calculated for the entire domain by our model using exponent values in the viscosity model (19) of $n = 1$ (Newtonian) and $n = 2$ (shear-thickening). Key features of the observed flow field, such as the increase in speed after the constriction in the northern branch and the stagnation at the channel split point, are present in both
Figures. The overall magnitude of the velocity – up to 9 m/s – is also in agreement with the observations.

## 5.4 Discussion

We evaluate the quality of the fit between the model and the observations by comparing the modelled speed with the observed speed for different power-law exponent ($n$) values, shown in Figure 7. Two areas of relatively large error are clear in the southern branch. We
attribute these mostly to uncertainties in the underlying topography data. We use a coarse pre-eruption DEM for an area where the overall slopes are very gradual (2-3 degrees). The mesh and model resolution is very high compared to the coarseness of the DEM (only 10 DEM grid points across the model), which can lead to inaccuracies in slope estimates. In addition, the DEM is from before the eruption, while the velocity data was captured a few
weeks after the channel was established. It is possible that by that time, some lava already deposited on the bottom of the channel and modified the topography.

An additional source of misfit could be due to the bottom stress calculation. We calculate

the thickness of the virtual layer over which the velocity transitions from the depth-integrated velocity to a value of zero (at the bottom boundary) via a two-layer model of vorticity. The two layers correspond to a mixed upper layer and a non-mixed bottom layer where the lava is loosing heat due to conduction. While this approach seems to be valid in terms or reproducing lava flow thicknesses observed by USGS surveys (USGS Hawaii Volcano Observatory, 2019) which place lava thicknesses between 5 m - 15 m, (alternative methods produce flows that are 3 times too thick), the two-layer model still is a simplification of reality that most likely introduces some errors.

Further, because we are limited to surface speed observations our error metrics will unavoidably have a misfit in them due to the fact that modeled speeds are depth-averaged. This effect will be small in regions of the channel where the Reynolds number is high because the lava speed will be more uniform over its thickness. However, in areas where the Reynolds number is low(er), model speeds will be less than observations. This is because the section of channel we modeled has minimal crust cover, and therefore, there is relatively zero stress at the top boundary of the lava flow meaning that lava speed should reach a maximum at the surface. This effect is evident in the northern portion of the northern channel and the portion of the southern channel between Latitude -154.886 °W and -154.8855 °W.

An interesting aspect of the model worth drawing attention to is how a change in the value of $n$ in the viscosity model affects numerical results. Figures 7 and 8 reveal that the overall fit improves with increasing $n$ values which corresponds to shear-thickening in our model. Similarly, the thicknesses predicted by our model for larger power-law exponent values (Figure 9) are closer to the range of thicknesses (5-15 m) measured by USGS survey (USGS Hawaii Volcano Observatory, 2019) and the apparent viscosity calculated by our model (shown in Figure 10) for large $n$ values is similar to rough estimates by the USGS (W. Thelen, pers. comm., 2018).

The lower error measures produced by our model for shear-thickening behaviour is a departure from other studies that find lava to behave as a shear-thinning fluid (e.g., Castruccio et al., 2010; Costa et al., 2009; Pinkerton, 1995). The disparity can most likely be attributed to differences in the particle content of the lava; in the studies of Castruccio et al. (2010), Costa et al. (2009), and Pinkerton (1995) the crystal/bubble content of the lava studied was low ($< 20\%$), whereas samples taken from the Fissure 8 flow during and after the eruption show a wide range of crystallinity and vesicularity, often with very high vesicle fraction of over 50% (Halverson et al., 2020). The improved overall fit of our model for higher $n$ values is most likely due to this high vesicle fraction and is consistent with the lab experiments of Smith (1997) and Lev et al. (2020) that show high bubble content can produce shear-thickening behavior. In fact, in the experiments of Sayag and Worster (2013), shear thickening behavior for a constant volume of fluid (as in our investigation) produce flow thicknesses that are less than the shear-thinning case, which is visible in our results in Figure 9. We conjecture that the increase in the viscosity is due to the fact that as the strain rate increases, the bubbles re-arrange in a fashion that makes it harder for the lava to flow. This effect should be especially pronounced in areas where the strain rate is high (think of the stress associated with solid boundaries in turbulence and how the vorticity created at these boundaries could cause bubbles to run into each other, impeding the flow). This effect is apparent in Figure 10 for the case $n = 2$, where the effective viscosity is low except in regions of high strain and high(er) Reynolds number, such as near the constriction

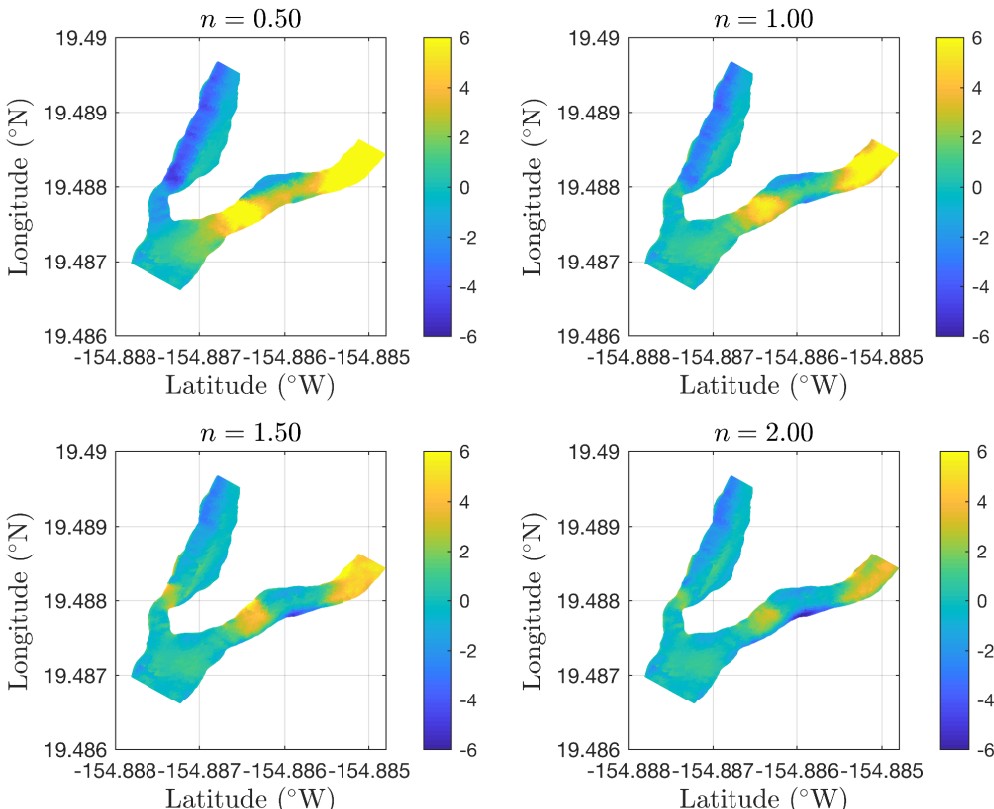

Figure 7: Map view of the difference between modeled lava speed and surface speed obtained from UAS video capture for various power law exponents. Colors represent difference in meters per second.

in the northern channel, at the channel walls where the slope is high in the southern channel, and at the bend area in the southern channel. In the future we will explore mathematical relationships that allow $n$ to be a function of the bubble/crystal content of the lava as well as examine the sensitivity of the model to non-Newtonian rheological parameters. We plan to infer the best fitting values for these parameters for a range of locations and times for the Fissure 8 lava flow.

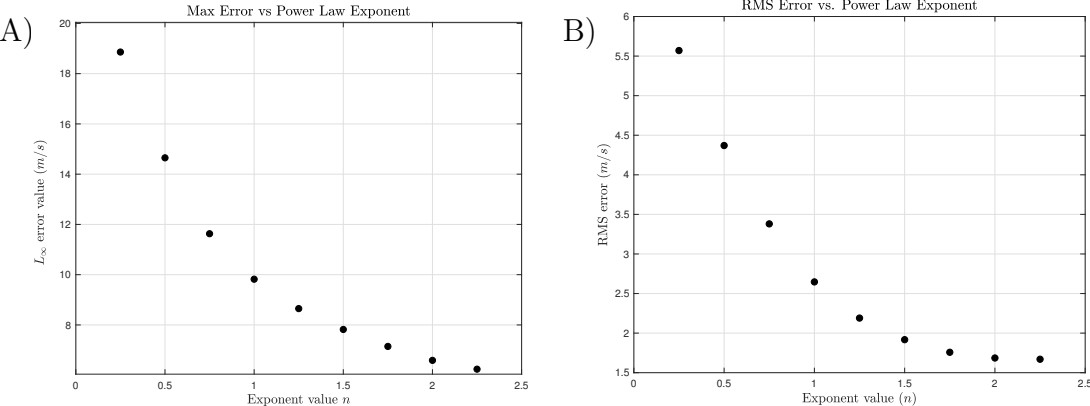

Figure 8: A) Maximum error in model speed versus power law exponent. B) Root mean square error in model speed versus power law exponent.

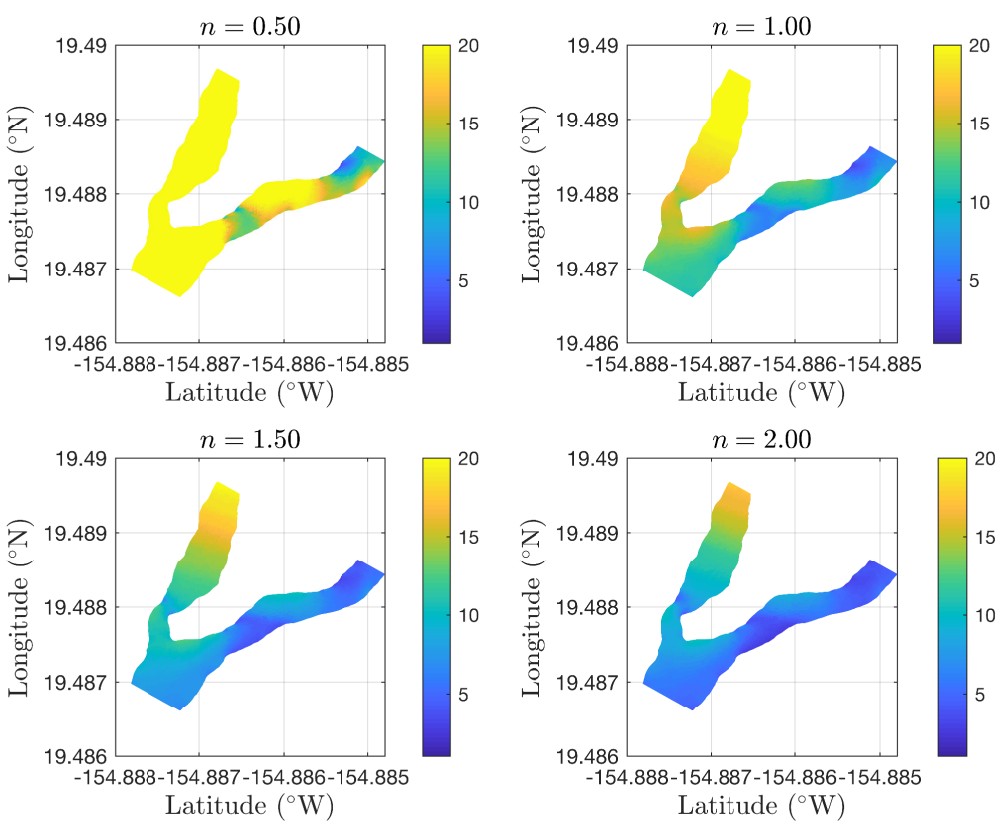

Figure 9: Map view of the modeled lava thickness for various power law exponents. Colors represent lava thickness in meters. Notice that the lava becomes thinner as the value of $n$ increases.

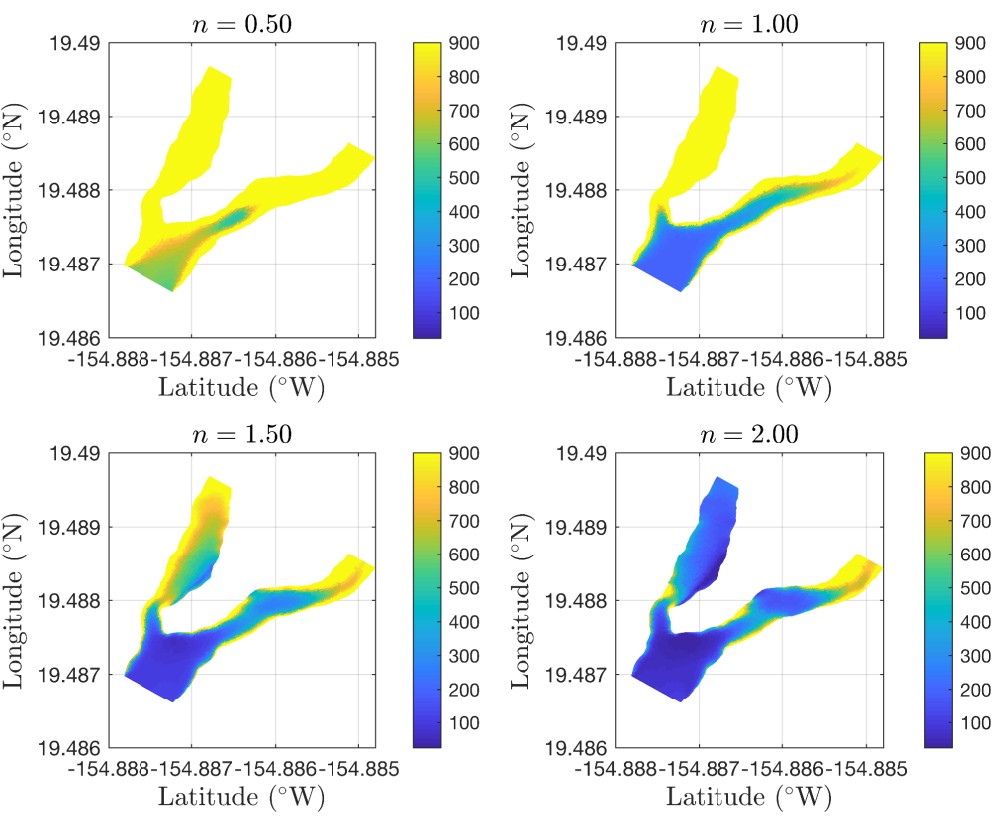

Figure 10: Map view of the effective modeled lava viscosity for various power law exponents. Colors represent viscosity in Pascal seconds.

# 6 Conclusions

We present a novel numerical model for quantifying high-speed lava flows in complex channels. The mathematical model consists of depth-averaged equations of mass, momentum, and energy and the equations are closed via a non-linear viscosity model. Because we use discontinuous Galerkin methods to discretize the mathematical model we are able to capture non-smooth transitions that can occur in lava flows, e.g. jumps in temperature, shear, and viscosity, see Figure 10. We overcome a major limitation to many depth-integrated models in terms of the need to use an adjustable friction coefficient by solving a heat transfer boundary layer problem coupled with a calculation for the thickness of a virtual layer over which the velocity transitions from the depth-integrated velocity to a value of zero (at the bottom boundary) via a two-layer model of vorticity. This novel approach results in lava flow thicknesses that are in the range of observed values as compared to simple linear schemes that produce flow thicknesses that are 3 times too thick. Further, the use of unstructured triangular meshes allows the model to accurately resolve complex braided channel systems that are commonly produced by basaltic lava flows. This was demonstrated on a section of the complex braided channel system that was created by the Fissure 8 flow from the 2018 Kilauea Lower East Rift Zone eruption with model results matching observational results quantitatively well. Future work will include using the new versatile model as a tool to infer lava properties and flux during volcanic crises.

## List of symbols

| Symbol | Name | Definition |
|---|---|---|
| $A$ | consistency constant | see §2.1 |
| $A_w$ | wetted area | $A_w = \mathrm{w}H$ |
| $\alpha_{rs}$ | Runge-Kutta coefficient | see §3.3 |
| $B$ | consistency constant | see §2.1 |
| $\mathbf{b}$ | discrete DG spatial matrix | see §3.3 |
| $\beta_{rs}$ | Runge-Kutta coefficient | see §3.3 |
| $C$ | consistency constant | see §2.1 |
| $dA$ | differential area | — |
| $dx_0, \ldots, dx_3$ | mesh size in verification test case | — |
| $\Delta t$ | numerical time step | — |
| $\delta_s \Delta t$ | sub time step in Runge-Kutta method | see §3.3 |
| $\boldsymbol{\delta}_z$ | length of virtual bottom boundary layer | $\boldsymbol{\delta}_z = (\delta_{zx}, \delta_{zy})'$ |
| $\epsilon$ | emissivity of lava | see Table 4 |
| $\mathbf{F}$ | flux function matrix | see equation (36) |
| $\hat{F}$ | numerical flux function | see equation (42) |
| $\mathbf{f}_b$ | body force vector | $\mathbf{f}_b = (g_x, g_y, g_z)'$ |
| $f_x$ | x-comp. of depth-integrated horizontal shear stress | $f_x = \int_{-h}^{\zeta} \frac{\partial \tau_{xx}}{\partial x} + \frac{\partial \tau_{yx}}{\partial y} \, dz$ |
| $f_y$ | y-comp. of depth-integrated horizontal shear stress | $f_y = \int_{-h}^{\zeta} \frac{\partial \tau_{xy}}{\partial x} + \frac{\partial \tau_{yy}}{\partial y} \, dz$ |
| $g_x, \ g_y, \ g_z$ | $x-, y-, z-$ components of gravitational acceleration | — |

| | | |
|---|---|---|
| $H$ | total depth of flow | $H = \zeta + h$ |
| $\mathcal{H}$ | Hilbert space | infinite dimensional space with finite energy |
| $h$ | steady reference depth of flow | see §2.3 |
| $h_b$ | depth of bottom thermal boundary layer | see §2.2 |
| $h_w$ | thickness of thermal boundary layer | — |
| $\mathbf{I}$ | identity matrix | $\begin{bmatrix} 1 & 0 & 0 & 0 \\ 0 & 1 & 0 & 0 \\ 0 & 0 & 1 & 0 \\ 0 & 0 & 0 & 1 \end{bmatrix}$ |
| $i$ | index counter for equations in (17) | $i = 1, \ldots, 4$ |
| $\hat{i}$ | unit vector in x-coordinate direction | — |
| $-\mathrm{i}$ | imaginary number | $-\mathrm{i} = \sqrt{-1}$ |
| $\mathbf{J}$ | Jacobian matrix | see §3.2.1 |
| $\mathbf{J}_x$ | Jacobian matrix in x-direction | see §3.2.1 |
| $\mathbf{J}_y$ | Jacobian matrix in y-direction | see §3.2.1 |
| $j$ | index counter for the set of elements in $\mathcal{T}_h$ | $j = 1, \ldots, \mathcal{N}$ |
| $\hat{j}$ | unit vector in y-coordinate direction | — |
| $\mathcal{K}$ | lava consistency | $Pa \cdot s$ |
| $\mathcal{K}_0$ | lava consistency constant | $s^{n-1}$ |
| $k$ | free surface wave number | $1/m$ |
| $\hat{k}$ | unit vector in z-coordinate direction | — |
| $k_b$ | thermal conductivity of ground | see Table 4 |
| $\tilde{k}_b$ | thermal conductivity of bottom boundary layer | see Table 4 |
| $k_c$ | convection constant | see Table 2 |

| | | |
|---|---|---|
| $k_T$ | heat transfer coefficient | |
| $L^2$ | $L^2 -$ error norm | $\lVert \cdot \rVert_2 \;=\; \left( \displaystyle\int_{\Omega_j} (\mathbf{U} - \mathbf{U}_h)^2 \; dA \right)^{1/2}$ |
| $\mathcal{L}_{hp}$ | right hand side of ODE | see §3.3 |
| $\mathbf{\Lambda}_{(\ )}$ | diagonal matrix of eigenvalues | see §3.2.1 |
| $\lambda$ | an eigenvalue of $\mathbf{J}$ | see §3.2.1 |
| $\lambda_{\max}$ | maximum eigenvalue of $\mathbf{J}$ | see §3.2.1 |
| $l$ | degree of freedom index for polynomial basis | see §3 |
| $\ell$ | degree of polynomial basis | see §3 |
| $\tilde{\mathbf{M}}$ | finite element mass matrix | see §3.3 |
| $m$ | number of eignvalues of $\mathbf{J}$ | see §3.2.1 |
| $\mu$ | viscosity | $Pa \cdot s$ |
| $\mathcal{N}$ | number of elements in $\mathcal{T}_h$ | — |
| $n$ | viscosity power law exponent | see §2.1 |
| $\hat{\mathbf{n}}$ | normal vector | vector perpendicular to a plane |
| $\Omega_j$ | element $j$ in $\mathcal{T}_h$ | — |
| $\boldsymbol{\omega}$ | vorticity | see equation (23) |
| $\omega$ | free surface wave frequency | 1/s |
| $P$ | pressure flux | $P = \frac{1}{2} g_z \left( H^2 - h^2 \right)$ |
| $\mathcal{P}$ | polynomial space for basis functions | see §3 |
| $P_w$ | wetted perimeter | $P_w = \mathrm{w} + 2H$ |
| $p$ | pressure | $Pa$ |
| $\partial \Omega$ | channel domain boundary | — |

| | | |
|---|---|---|
| $\partial\Omega_j$ | boundary of element $j$ in $\mathcal{T}_h$ | — |
| $\Pi_h$ | slope limiter | see Cockburn and Shu (2001) |
| $\phi$ | basis functions | combinations of Jacobi polynomials |
| $\psi$ | test function | $\psi \in \mathcal{V}$ |
| $\psi_h$ | finite element approximation of $\psi$ | see equation (40) |
| $\psi_l$ | polynomial coefficients of $\psi_h$ | see equation (40) |
| $\mathbf{Q}_{\text{in}}$ | channel inflow flux | $m^3/s$ |
| $\mathbf{q}$ | heat flux | $\mathbf{q} = \left( k_T \dfrac{\partial T}{\partial x}, k_T \dfrac{\partial T}{\partial y}, k_T \dfrac{\partial T}{\partial z} \right)'$ |
| $\dot{\mathbf{q}}$ | internal heat generation/dissipation | — |
| $\dot{\bar{q}}$ | internal heat generation/dissipation | — |
| $q_b$ | bottom heat flux | see §2.2 |
| $\bar{q}_i$ | depth-integrated internal heat flux | — |
| $q_s$ | surface heat flux | see §2.2 |
| $\mathbf{R}$ | vector of DG spatial operator | see §3.3 |
| $\mathcal{R}_{(\ )}$ | Matrix of right eigenvectors of $\mathbf{J}$ | see §3.2.1 |
| $\mathcal{R}_{(\ )}^{-1}$ | Matrix of left eigenvectors of $\mathbf{J}$ | see §3.2.1 |
| $Re$ | Reynold's number | $Re = \dfrac{\rho\lVert\mathbf{u}\rVert A_w}{\mu P_w}$ |
| $\mathbf{r}$ | eigenvector of $\mathbf{J}$ | see §3.2.1 |
| $\rho$ | density | $Kg/m^3$ |
| $r,\ s$ | index counters in Runge-Kutta time stepper | — |
| $\mathbf{S}$ | source vector | see equation (36) |
| $\mathcal{S}$ | number of Runge-Kutta stages | — |

| | | |
|---|---|---|
| sgn() | sign of argument | — |
| $\sigma_b$ | Stefan-Boltzmann constant | see Table 4 |
| $\boldsymbol{\sigma}_{\text{wall}}$ | normal stress vector at wall | $(-\hat{\mathbf{n}}\boldsymbol{\tau})'$ |
| $T$ | temperature | K |
| $\bar{T}$ | depth averaged temperature | $\bar{T} = \dfrac{1}{H}\displaystyle\int_{-h}^{\zeta} T\ dz$ |
| $\hat{T}$ | amplitude of $\bar{T}$ in verification test | see §4 |
| $\mathcal{T}_h$ | finite element triangulation of $\Omega$ | $\mathcal{T}_h = \{\Omega_j\}_{j=1}^{\mathcal{N}}$ |
| $T_{\text{atm}}$ | atmospheric temperature | K |
| $T_{\text{ground}}$ | ground temperature | K |
| $t$ | time | $s$ |
| $\hat{\mathbf{t}}$ | tangential vector | vector parallel to a plane |
| $t_i$ | initial time in averaging window | $s$ |
| $t_f$ | final time in averaging window | $s$ |
| $\boldsymbol{\tau}$ | stress tensor | see §2 |
| $\boldsymbol{\tau}_b$ | bottom stress vector | $\boldsymbol{\tau}_b = (\tau_{bx}, \tau_{by})'$ |
| $\boldsymbol{\tau}_s$ | surface stress vector | $\boldsymbol{\tau}_s = (\tau_{sx}, \tau_{sy})'$ |
| $\boldsymbol{\sigma}_{\text{wall}}$ | normal stress vector at wall | $(-\hat{\mathbf{n}}\boldsymbol{\tau})'$ |
| $(\ )'$ | transpose operator | $(u, v, w)' = \begin{bmatrix} u \\ v \\ w \end{bmatrix}$ |
| $\mathbf{U}$ | solution vector | see equation (36) |
| $\tilde{\mathbf{U}}$ | vector of polynomial coefficients (degrees of freedom) | see §3.3 |
| $U_h$ | finite element approximation of U | see equation (39) |

| | | |
|---|---|---|
| $U_l$ | polynomial coefficients of $U_h$ | see equation (39) |
| $\mathbf{u}$ | depth dependent velocity | $\mathbf{u} = (u, v, w)^{'}$ |
| $\|\mathbf{u}\|$ | magnitude of velocity | $\|\mathbf{u}\| = (u^2 + v^2 + w^2)^{1/2}$ |
| $\bar{\mathbf{u}}$ | depth-averaged velocity vector | $\bar{\mathbf{u}} = \left( \dfrac{1}{H} \int_{-h}^{\zeta} u \, dz, \ \dfrac{1}{H} \int_{-h}^{\zeta} v \, dz \right)^{'}$ |
| $u$ | $x-$ component of velocity | $u = dx/dt$ |
| $\hat{u}$ | amplitude of $\bar{u}$ in verification test | see §4 |
| $\mathcal{V}$ | admissible space of functions | $\mathcal{V} \in \mathcal{H}$ |
| $\mathcal{V}_{hp}$ | finite dimensional subspace | $\mathcal{V}_h \in \mathcal{V}$ |
| $v$ | $y-$ component of velocity | $v = dy/dt$ |
| $\hat{v}$ | amplitude of $\bar{v}$ in verification test | see §4 |
| $w$ | $z-$ component of velocity | $w = dz/dt$ |
| $\bar{w}$ | measure of depth averaged vertical velocity | see equation (25) |
| $\tilde{\mathbf{w}}$ | right hand side function in RK method | see §3.3 |
| $\mathrm{w}_{\mathrm{in}}$ | channel width at inlet | — |
| $\mathbf{x}$ | Cartesian coordinate | $\mathbf{x} = (x, y, z)^{'}$ |
| $x_0, \ x_L$ | x-coordinate of inlet and outlet boundaries | — |
| $y_0, \ y_L$ | y-coordinate of channel walls | — |
| $\tilde{z}$ | thermal boundary layer coordinate | $\tilde{z} \in [0, -h_b]$ |
| $z_b$ | z-coordinate where thermal boundary layer begins | see §2.2 |
| $\zeta$ | free surface elevation | $\zeta = H - h$ |
| $\hat{\zeta}$ | amplitude of surface elevation in verification test | see §4 |
| $\tau_{\mathrm{yield}}$ | yield strength of lava | — |

| | | |
|---|---|---|
| $\nabla$ | gradient operator | $\nabla = \left(\dfrac{\partial}{\partial x}, \dfrac{\partial}{\partial y}, \dfrac{\partial}{\partial z}\right)'$ |
| $\bar{\nabla}$ | horizontal gradient operator | $\bar{\nabla} = \left(\dfrac{\partial}{\partial x}, \dfrac{\partial}{\partial y}\right)'$ |
| $\mathbf{0}$ | vector of zeros | $(0, \dots, 0)'$ |

## Computer code and data

The computer code and data used in this investigation can be found at `https://zenodo.org/badge/latestdoi/267726380`.

## Author contribution

CJC developed the model code and performed all model simulations. EL assisted in the Unoccupied Aerial System (UAS) data collection and utilized the Optical Flow technique to quantify lava velocities from the UAS data. CJC prepared the manuscript with contributions from EL.

## Competing interests

The authors declare that they have no conflict of interest.

## Financial support

This investigation was supported by the National Science Foundation under Grant No. EAR-1654588.

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
