# Peer review of "A discontinuous Galerkin finite element model for fast channelized lava flows v1.0"

_Geoscientific Model Development, 2020_

## Referee Comment (RC1) · Anonymous Referee #1 · 6 Nov 2020

The paper presents a numerical model that solves the thermodynamical problem of lava flowing in channels of complex geometry. The paper presents the differential equations of the problem, the time and space discretization, the numerical tests, the application to a real case study and release the Matlab code and related files. The paper is well-structured but the exposition is not rigorous and I found the reading not fluid and suitable only for very specialized readers. Moreover in the text you come across changes in the names of the functions/parameters used, and often authors use the same letter to indicate different parameters ($h$ is for mesh and height, $p$ is for polynomial degree and pressure..). This makes reading the article very annoying. Nevertheless, I suggest the publication of the paper after minor revision. In the following, my comments. It is not clear whether the proposed method is an adaptation

of other methods developed in the fields of atmospheric fluid dynamics or is an original method developed "ad hoc" on the problem of lava flows. The works of Kubakto et al., (2006; 2014; 2015) are repeatedly cited and the origin of continuous and discrete equations is not well distinguishable. I ask the authors to make an effort to clarify this distinction also by inserting sentences and more references, in the text to guide the reader, i.e. explanations and references before the demonstrative sections. About that, I hope I don't have misinterpreted it, but it seems to me that the layout of the work is inspired by oceanographic studies. In this sense, the modeling of the variations in height of the free surface applied to lava flow is very interesting. Obviously, what distinguishes an oceanographic fluid dynamics problem from a volcanological one is the complexity lava rheology and its dependence by temperature, bubble, crystal content and composition. Lava rheology is generally considered non-newtonian and in particular shear thinning or pseudoplastic and the authors include in eq. (3) a non-linear rheology in their modeling. The dependence of viscosity on velocity gradients is expressed by means of the fluid consistency coefficient, modeled following Giordano et al., (2008); the power-law index, on the other hand, does not have an analytical expression. In the application to a real case, the power-law index turns out to be an accommodation parameter whose variations significantly influence the final model. Therefore, the use of a VFT model for consistency with a constant and arbitrary varying n-index should be justified in some way, maybe adding some references. Laboratory experiments showed that n-index is not constant but can vary with temperature too (e.g. Sonder et al., 2006) and authors should add comments on this aspect in section 2.1. The model verification is achieved by using manufactured solution method but from the tables (from 1 to 3) it is not clear how P0, P1, P2 are defined (also in this case P0 sometimes is expressed with an index some others with a pedis). Also in this case author should justify the choice of exponential functions for their tests or add references. The results applied to a real case should show also the element size of the chosen mesh to give the idea of the errors in the final model. Then, the authors should explain if the errors in the final model at the boundaries and

at the base of the domain are almost an order less than the errors introduced by the variations of the DEM, in order not to risk making the use of the complex geometry of the channel useless. This aspect should be discussed in the text. Finally, I suggest the authors a re-reading of their manuscript with the aim of clarifying these aspects, adding more references, adjust the nomenclature of functions/parameters and simplify the exposition.

Please also note the supplement to this comment:
https://gmd.copernicus.org/preprints/gmd-2020-184/gmd-2020-184-RC1-supplement.pdf

---

## Referee Comment (RC2) · Anonymous Referee #2 · 7 Nov 2020

In this study, the authors present a numerical model based on discontinuous Galerkin finite element method and apply it to channels with complex geometry: the space discretization is obtained by an unstructured triangular mesh, the time discretization by the Runge-Kutta method, the numerical tests via the manufactured solutions method. Moreover, the authors model high-speed lava flow and overcome the modeling of turbulence by a two-layer model of vorticity and introducing a stress term at the boundary of the domain. The authors present an application to the Kilauea 2018 eruption and retrieve flow velocity and rheology. The manuscript is a model description paper, but the description of mathematical formalism needs corrections. The manuscript is not well-organized in sections and some changes are needed before publication. (1) In the title: what v1.0 refer to? I didn't find any explanation for this alphanumeric symbol.

Is this the first version of their software? (2)I found some problems with references. At line 21 authors refer to Neal et al., (2019) to find the value of lava speed of 11 m/s but I didn't find this number in the cited work. They also refer to Re>3000, where they found this information is not explained. (3)In the introduction section (lines 59-75) the method is exposed in detail and this part should be moved in the method section. The whole introduction is short, and references are not enough to introduce their method. The method is novel, it is a combination of many different techniques and so, each of them should be introduced with accurate references in the introduction section, explaining the need to use the chosen methodology. Methodologies that are commonly used for hurricane storms are a novelty for lava flows and should be described better. (4) In the mathematical model the authors do not justify the choice of neglecting viscous dissipation. The effect of viscous dissipation is discussed in other numerical studies (Costa and Macedonio, 2003; Cordonnier et al., 2012) and the effect is to dramatically change the velocity of lava flow at the boundaries, introducing local vorticities and increasing the Reynold's number. Please discuss these limitations. (5) Lines 87-99: this part should be extended because it is unreadable as it is. Formulae need space and explanations. If they are not necessary, remove them. (6) Figure 2 doesn't contain all the geometrical and physical parameters used in the mathematical model. Add all the parameters (v, H, x, y....) in Figure 2. (7) Lines 102-104: in this section the upper and bottom surface boundary condition should be introduced here and not divided in different sub-sections. If authors prefer a different sub-section, they can name it "boundary conditions" instead of "stress term" describing the stress term at the bottom and the moving surface at the top. (8) Fluid viscosity function is divided in two sections and should not be. The choice of the rheology model of Giordano et al (2008) should be motivated. Authors claim in their abstract the they will use a non-linear viscosity function but, in the text, they show a temperature dependent viscosity function. The used viscosity model can be considered non-linear respect to the energy equation and not to the dynamic equations. In this work, non-linearity is obtained by a varying exponent (n-1). How the power-law exponent is chosen in the final model?

There are not many works on power-law rheology experiments that model both the exponent n and the consistency k of the fluid, and their variation with temperature and composition. Starting from the pioneering work of Sonder et al., (2006), Hobiger et al., (2011) developed an interesting model for basaltic lava used for both analytical and numerical modelling of lava flows (Tallarico et al, 2011; Filippucci et al., 2017). It would be more correct if, in the presentation of their solving method, the authors described a temperature-dependent viscosity with the possibility of introducing a non-linearity in the equations by appropriately choosing a constant value of the exponent of the power law. The authors should discuss this limitation in the discussion section and in the description of the viscosity funtion. (9) Eq (4) what is C'? is it a refuse? (10) Lines 177-179: please rewrite the sentence. (11) Given the huge number of different parameters used in their sections, I think that a table of acronyms and parameters would be helpful for reviewers and for readers. For example: QR and Qc in (20) are not defined before but I can imagine that are R stands for radiative and C for conductive heat, is it right? Please, use a unique name for functions. Another example in lines: 220-225: what j rapresents? and the pedix e? Formulae are confused and it seems that they came from a patchwork of other works. The mathematical formalism does not seem to have been written specifically for this article. (12) Model results are very interesting, but the problem of the chosen rheology is evident in lines from 410, where authors describe limitations that I appreciated. (13) It would be interesting if in the numerical verification section, authors add the time costs of each computation also as order of magnitude (minutes, hours, days..). The same information is useful also in the results section for explain the choice of the mesh dimension. Is the non-linear problem more expensive than the newtonian one?

---

## Author Comment (AC1) · 21 Jan 2021

Response to referee comments –

Reviewer #1 1. The paper is well-structured but the exposition is not rigorous and I found the reading not fluid and suitable only for very specialized readers.

The Authors agree that the initial draft of the manuscript was more suitable for specialized readers. To mitigate this problem we have added material to the introduction to streamline the presentation of material for a broader audience, and expanded the mathematical model section to make it more rigorous.

2. Moreover in the text you come across changes in the names of the functions/parameters used, and often authors use the same letter to indicate different parameters (h is for mesh and height, p is for polynomial degree and pressure..).

The Authors would like to thank the Reviewer for pointing this out. The work presented in the manuscript is at the intersection of volcanology, numerical methods, and optical analysis meaning at times there have been crossover in symbols commonly used by each field. We have revised the manuscript to ensure that this crossover no longer exists. We have also added a table of symbols at the end of the manuscript along with each symbol's definition.

3. It is not clear whether the proposed method is an adaptation of other methods developed in the fields of atmospheric fluid dynamics or is an original method developed "ad hoc" on the problem of lava flows. The works of Kubakto et al., (2006; 2014; 2015) are repeatedly cited and the origin of continuous and discrete equations is not well distinguishable. I ask the authors to make an effort to clarify this distinction also by inserting sentences and more references, in the text to guide the reader, i.e. explanations and references before the demonstrative sections.

The inspiration for the dynamical model was the depth integrated models of Kubatko et al. and Dawson et al., however, the development of the thermodynamic model and the coupling of the two models via the bottom stress term was developed specifically for the problem of lava flows. We have updated the manuscript to make this point clear to the reader.

4. It seems to me that the layout of the work is inspired by oceanographic studies. In this sense, the modeling of the variations in height of the free surface applied to lava flow is very interesting. Obviously, what distinguishes an oceanographic fluid dynamics problem from a volcanological one is the complex lava rheology and its dependence by temperature, bubble, crystal content and composition. Lava rheology is generally considered non-newtonian and in particular shear thinning or pseudoplastic and the authors include in eq. (3) a non-linear rheology in their modeling. The dependence of viscosity on velocity gradients is expressed by means of the fluid consistency coefficient, modeled following Giordano et al., (2008); the power-law index, on the other hand, does not have an analytical expression. In the application to a real case, the power-law index turns out to be an accommodation parameter whose variations significantly influence the final model. Therefore, the use of a VFT model for consistency with a constant and arbitrary varying n-index should be justified in some way, maybe adding some references.

The power law exponent in the model does not have an analytic expression, but this is common in volcanology due to the complexity of the materials involved. Most expressions for the dependency of the power-exponent n on variables such as temperature or crystal content are developed empirically through experiments. We point that the expression for the consistency (i.e., its dependence on temperature through the Giordano et al model) is also based on three constants (A, B, C) that are found empirically for each specific composition. We treat n similarly. Our goal in this work is to similarly find values of n that allow the model to replicate the field observations; thus, an analytical expression for n is not needed. By varying the exponent we are still able to ascertain information about the lava flow, mainly, did a high bubble content make the lava behave as a shear thickening fluid or a shear thinning fluid? In the specific application of the model to the Kilauea lava flows the high vesicularity of the lava displayed a shear thickening behavior. That said, we are working on incorporating an analytical expression for three-phase rheology based on laboratory experiments (Birnbaum et al., in review in Geology) into future releases of the model.

5. Laboratory experiments showed that n-index is not constant but can vary with temperature too (e.g. Sonder et al., 2006) and authors should add comments on this aspect in section 2.1.

Sonder et al 2006 in fact show that n varies much less with temperature compared with the temperature dependence of the consistency (K in their formulation) is much greater ( K = 59.9 Pa s, n = 0.563 at 1473 K, and K = 17.4 Pa s, n = 0.618 at 1498 K).

6. The model verification is achieved by using a manufactured solution method but from the tables (from 1 to 3) it is not clear how P0, P1, P2 are defined (also in this case P0 sometimes is expressed with an index, some others with a pedis).

P0 corresponds to a constant basis function, P1 corresponds to a linear basis function, and P2 corresponds to a quadratic basis function. We have updated the text to make this clear to the reader as well as made our mathematical notation consistent.

7. Also in this case the author should justify the choice of exponential functions for their tests or add references.

We have added our justification to the text which is based on the idea of a lava source term that effuses lava in sinusoidal pulsations.

8. The results applied to a real case should also show the element size of the chosen mesh to give the idea of the errors in the final model.

We have included the element size of the mesh in the revised manuscript.

9. Then, the authors should explain if the errors in the final model at the boundaries and at the base of the domain are almost an order less than the errors introduced by the variations of the DEM, in order not to risk making the use of the complex geometry of the channel useless. This aspect should be discussed in the text.

The side wall boundaries of the model domain were picked from the aerial videos used to measure the flow surface velocity field, and their location is therefore relatively accurate. In contrast, the pre-eruption digital elevation data had a lower spatial resolution of approximately 90 m/pixel, making the shape of the bottom boundary less accurate and thus a larger source of error. Resolving the complex geometry of the channel is important to reproduce the lava flow jet formed by the constriction in the northern portion of the lava channel.

10. Finally, I suggest the authors a re-reading of their manuscript with the aim of clarifying these aspects, adding more references, adjust the nomenclature of functions/parameters and simplify the exposition

The Authors have re-read the manuscript and addressed all of the Reviewers comments and we would like to thank the Reviewer for the comments which have helped strengthen the manuscript.

Reviewer #2 1. In the title: what v1.0 refer to? I didn't find any explanation for this alphanumeric symbol. Is this the first version of their software?

V1.0 stands for version 1.0 of the software. We have made this clear to the reader in the revised version of the manuscript (GMD requires a unique alphanumeric identifier for model description papers).

2. At line 21 authors refer to Neal et al., (2019) to find the value of lava speed of 11 m/s but I didn't find this number in the cited work. They also refer to Re>3000, where they found this information is not explained.

The reference should have been to Patrick et al., 2019, where there are time series of lava velocities reaching 12-14 m/s during peak times near the vent, and slightly lower speeds near the focus site for our example. Reynolds number is calculated using the standard formula, now given in the symbols table. Using values for the 2018 flows, i.e., a nominal viscosity of $\sim$100 Pa s, a channel depth of 10m, a velocity of 15 m/s and a density of 2000 kg/m3, gives a Reynolds number of 3000.

3. In the introduction section (lines 59-75) the method is exposed in detail and this part should be moved in the method sec-tion. The whole introduction is short, and references are not enough to introduce their method. The method is novel, it is a combination of many different techniques and so, each of them should be introduced with accurate references in the introduction section, explaining the need to use the chosen methodology. Methodologies that are commonly used for hurricane storms are a novelty for lava flows and should be described better.

The Authors have added a significant amount of material and references to the introduction to explain the need for the methodology presented in the manuscript. In addition, the introduction was split into subsections to help the reader navigate the different aspects of the material.

4. In the mathematical model the authors do not justify the choice of neglecting viscous dissipation. The effect of viscous dissipation is discussed in other numerical studies (Costa and Macedonio, 2003; Cordonnier et al., 2012) and the effect is to dramatically change the velocity of lava flow at the boundaries, introducing local vorticities and increasing the Reynold's number. Please discuss these limitations.

The model has the ability to handle viscous dissipation, however, we did not include this effect for the Kilauea lava flows because the low viscosity and high temperature make heating by viscous dissipation small relative to heat loss through radiation and conduction. For the Kilauea flows, radiation is almost three orders of magnitude larger than viscous dissipation ($\sim$7e4 vs $\sim$1e2 W/m), and one order of magnitude larger than conduction ($\sim$1e3 W/m). In addition, we point that the model of Costa and Macedonio deals with flow inside a closed tube, where radiation is not relevant; they also show that viscous dissipation becomes important only at high Nahme number values.

5. Lines 87-99: this part should be extended because it is unreadable as it is. Formulae need space and explanations. If they are not necessary, remove them. We have streamlined the presentation of the mathematical model to make it more rigorous and easy to understand for the reader. Figure 2 doesn't contain all the geometrical and physical parameters used in the mathematical model. Add all the parameters (v, H, x, y. . ..) in Figure 2.

We have updated Figure 2 to contain all of the parameters.

6. Lines 102-104: in this section the upper and bottom surface boundary condition should be introduced here and not divided in different sub-sections. If authors prefer a different sub-section, they can name it "boundary conditions" instead of "stress term" describing the stress term at the bottom and the moving surface at the top.

In depth-integrated models there is not a true bottom boundary condition because the vertical profile of the mathematical model is eliminated via integration and the bottom stress term takes the form of a source term in the momentum equations. To make this more clear to the reader, we have updated the manuscript and begin from the full three-dimensional model with boundary conditions and derive the depth-integrated model with source terms that account for the surface and bottom stress as well as the horizontal boundary conditions.

7. Fluid viscosity function is divided in two sections and should not be. The choice of the rheology model of Giordano et al (2008) should be motivated. Authors claim in their abstract that they will use a non-linear viscosity function but, in the text, they show a temperature dependent viscosity function. The used viscosity model can be considered non-linear with respect to the energy equation and not to the dynamic equations. In this work, non-linearity is obtained by a varying exponent (n-1). How the power-law exponent is chosen in the final model?

The authors believe this comment stems from a confusion in terminology. The term 'non-linear' here is used to refer to the power-law dependence of viscosity on the strain rate (Eqn. 19), and not to the non-linearity of the system of equations themselves in the mathematical sense. The Authors have updated the text to elucidate the non-linear rheology model: The viscosity of the model is dependent on strain gradients in addition to the temperature. In the current version of the model the power law exponent is varied as an input parameter and compared to observational values, providing the user with insight into the rheology of the lava. However, future releases of the model will make use of a three phase power law rheology model based on the laboratory experiments by Birnbaum et al., currently in review in Geology.

8. There are not many works on power-law rheology experiments that model both the exponent n and the consistency k of the fluid, and their variation with temperature and composition. Starting from the pioneering work of Sonder et al., (2006), Hobiger et al., (2011) developed an interesting model for basaltic lava used for both analytical and

numerical modelling of lava flows (Tallarico et al, 2011; Filippucci et al., 2017). It would be more correct if, in the presentation of their solving method, the authors described a temperature-dependent viscosity with the possibility of introducing a non-linearity in the equations by appropriately choosing a constant value of the exponent of the power law. The authors should discuss this limitation in the discussion section and in the description of the viscosity function.

The Authors have updated the text detailing the limitations of the current viscosity model.

9. In Eq (4) what is C'? is it a refuse?

This was a comma after the equation, which unfortunately looked like a tag on the C. The authors removed the comma to avoid confusion.

10. Lines 177- 179: please rewrite the sentence.

The Authors have rewritten the text.

11. Given the huge number of different parameters used in their sections, I think that a table of acronyms and parameters would be helpful for reviewers and for readers. For example: QR and Qc in equation (20) are not defined before but I can imagine that are R stands for radiative and C for conductive heat, is it right? Please, use a unique name for functions. Another example in lines: 220-225: what j represents? and the pedix e? Formulae are confused and it seems that they came from a patchwork of other works. The mathematical formalism does not seem to have been written specifically for this article.

The mathematical formalism is applicable to any system of hyperbolic or parabolic partial differential equations. We have reread the manuscript and updated it to ensure that all of the nomenclature is consistent and defined within the text.

12. Model results are very interesting, but the problem of the chosen rheology is evident in lines from 410, where authors describe limitations that I appreciated.

[Figure]

13. It would be interesting if in the numerical verification section, authors add the time costs of each computation also as order of magnitude (minutes, hours, days..). The same information is useful also in the results section for explaining the choice of the mesh dimension. Is the non-linear problem more expensive than the newtonian one?

The Authors have added a discussion in terms of the cost of computation for different polynomial approximations used in the verification section and we have added an explanation for the choice of the mesh size used (which is determined by the mesh generator based on the topography and topology of the lava channel domain). There is no difference in terms of computational cost between the non-linear viscosity and the Newtonian viscosity because we use an explicit time stepping scheme.

―――――――――――――――――――――